# The Microphysics of Clouds over the Antarctic Peninsula – Part 1: Observations

Tom Lachlan-Cope[1], Constantino Listowski[1], Sebastian O'Shea[2]

[1] British Antarctic Survey, NERC, High Cross, Madingley Rd, Cambridge CB3 0ET, UK

[2] School of Earth, Atmospheric and Environmental Sciences, University of Manchester, Oxford Road, Manchester, M13 9PL, UK

*Correspondence to*: Tom Lachlan-Cope (tlc@bas.ac.uk)

**Abstract.** Observations of clouds over the Antarctic Peninsula during summer 2010 and 2011 are presented here. The Peninsula is up to 2,500m high and acts as barrier to weather systems approaching from the Pacific sector of the Southern Ocean. Observations of the number of ice and liquid particles as well as the ice water content and liquid water content in the clouds from both sides of Peninsula and from both years were compared. In 2011 there were significantly more water drops and ice crystals, particularly in the east where there were approximately twice the number of drops and ice crystals in 2011.

Ice crystals observations as compared to Ice Nuclei parameterisation suggest that secondary ice multiplication at temperatures around -5°C is important for ice crystal formation on both sides of the Peninsula below 2000 meters. Also, back trajectories have shown that in 2011 the air masses over the Peninsula were more likely to have passed close to the surface over the sea ice in the Weddell Sea. This suggests that the sea ice covered Weddell Sea can act as a source of both cloud condensation nuclei and ice nuclei.

## 1 Introduction

There have been very few in situ measurements of cloud microphysical properties over the Antarctic Continent (Bromwich et al., 2012; Lachlan-Cope, 2010). However, there is evidence, from surface radiation measurements, that clouds are poorly represented within numerical models over Antarctica (King et al., 2015; Bromwich et al., 2013) and over the surrounding oceans (Flato et al., 2013; Bodas-Salcedo et al., 2014). To correct these errors in climate (and forecast) models a better understanding of the microphysical processes controlling these clouds is needed. In situ observations of cloud and aerosol properties over the Antarctic Continent are required to develop and validate model parameterisations of these clouds. This paper presents observations that start to address this issue.

The main part of the Antarctic Continent is an ice sheet that rises to over 4000 meters above sea level (asl). Coming off this continental mass and heading north towards South America is the Antarctic Peninsula (see Fig. 1). The Antarctic Peninsula is less than 100km wide for the most part and rises to over 3000m in places. Although isolated measurements have been made over the main continent (Belosi et al., 2014) more measurements have been made over the Peninsula. Measurements of Ice

Nucleating Particles (INP) have been made at Palmer Station (64°46′27″S 64°03′11″W) (Saxena, 1983) and direct measurements have been made from the surface of cloud particle phase and size from a field camp on the spine of the Peninsula (Lachlan-Cope et al., 2001). During the southern summer in 2010 and 2011 measurements of basic meteorological parameters, turbulence and cloud microphysical properties were taken using the British Antarctic Survey's instrumented Twin Otter aircraft and it is these measurements that are considered here. Some of these data from four flights in 2010 have already been published (Grosvenor et al., 2012) but that work concentrated on the ice crystals present while here we consider both the liquid and ice present in the clouds in 24 flights during the two campaigns. Results are presented on the number of liquid drops and ice particles present in the clouds as well as the Liquid Water Content (LWC) and Ice Water Content (IWC).

As observations were made on both sides of the Antarctic Peninsula there is an opportunity to see if the cloud microphysical properties vary from one side to the other. The western side of the Peninsula is exposed to Southern Ocean that, in the summer at least, is relatively ice free. However, on its eastern side, the Peninsula is bordered by the western part of the Weddell Sea, which remains largely ice covered for most of the year. If the main source of Cloud Condensation Nuclei (CCN) were from the ocean surface it would be expected that the total number of liquid droplets would be different from one side to the other. This hypothesis is supported by results from the GOCART simulations, see Fig. 1 of Thompson and Eidhammer (2014), which show sharp discontinuity in sea salt aerosols across the Antarctic Peninsula in February, with concentrations on the western side at least twice as large as on the eastern side. However the GOCART simulations do not include sources of aerosol within the sea ice pack that have been suggested by some authors (Yang et al., 2008).

This paper is organised as follows: in section 2 the observations obtained from the two aircraft campaigns are presented. The results from both years and both sides of the Antarctic Peninsula for liquid droplets, ice crystals, and aerosols are analysed in section 3. In section 4 the results are discussed and suggestions are made for the most plausible explanations of the temporal and regional differences observed in clouds and aerosols across the Peninsula. Section 5 summarizes the findings and concludes on the possible implications of the results.

Part II of this paper will look at the application of these observations to numerical modelling.

**2. Observations**

**2.1 Aircraft Measurements**

Two airborne field campaigns were performed during February and March 2010, and January and February 2011, based at Rothera Research Station (67° 34' S, 68 ° 08' W) on the Antarctic Peninsula. During these two periods, flights were made to study a variety of meteorological phenomena including boundary layer (Fiedler et al., 2010), orographic flow (Elvidge et al., 2014) and cloud studies. A total of 64 flights were completed during these periods. This paper mainly considers 24 of these

flights (12 per year) where detailed cloud microphysical measurements were collected. Fig. 1 shows the flight tracks of these 24 cloud flights over the two periods. The sampling took place on both sides of the Peninsula flying between 61° and 73° W. The predominant cloud types by far were stratus or altostratus, normally in multiple thin layers. A cross section, across the Peninsula, through the flights is shown in Fig. 2. It should be noticed that the flights in 2010 did not go as far west and sampling
on the west side of the Peninsula was largely limited to altitudes below 2000 m west of 69° W.

The observations were made with the British Antarctic Surveys instrumented Twin Otter aircraft (see King et al., 2008). This aircraft is fitted with a variety of instruments to measure temperature, humidity, radiation, turbulence and surface temperature. The aircraft was also fitted with a Droplet Measurements Technology Cloud Aerosol and Precipitation Spectrometer (CAPS) (Baumgardner et al., 2001) carried on a wing-mounted pylon. The CAPS probe can measure cloud particle diameters from 0.5
to 50 μm and can image large particle from 25 μm to 1.5 mm.

The CAPS instrument contains three discrete instruments: The Cloud and Aerosol Spectrometer (CAS) which measures the diameter of particles between 0.5 to 50 μm at a frequency of one Herz. While the CAS used in this campaign did not have a full anti shatter inlet some modifications had been made to reduce the effect of shattering on the inlet by removing the shroud that was originally fitted to the inlet. The Cloud Imaging Probe (CIP) images particles between a diameter of 25μm and 1.5mm,
with 25 μm pixel resolution and had not at the time of this campaign been fitted with anti-shatter tips. However, a study of the particle inter arrival times indicated very few shattered particle and these were removed by eliminating particles that arrive within 1μs. A hotwire Liquid Water Content sensor (LWC) which measures the cloud liquid water was also fitted. This study only uses data from the CAS and CIP, although the Hotwire was used to help validate the CAS data.

A previous study (Grosvenor et al., 2012) has already looked at a small subset of this data and reported errors with the data
from the CAS instrument. In particular, it appeared to be over counting when integrated water content from the CAS was compared with the LWC sensor. After investigation this was found to be due to the air accelerating in the tube of the CAS instrument. Studies in the Cambridge University Markham wind tunnel using a fine pitot tube to measure the speed up within the tube showed an increase that would result in the count being increased by 1.47 and this has been accounted for in this latest study. When this correction is applied the liquid water content calculated by integrating the CAS data for most flights agree
within 15% with the hotwire LWC sensor – although the hotwire sensor tends to under read at high values of LWC.

The CIP instrument produces shadow images of the larger cloud particles and small precipitation size particles onto a CCD array. Data processing is performed on these images to derive size segregated ice crystal and large liquid drop number concentrations. Particles that are imaged by the extreme ends of the CCD array are rejected and this means that the effective

collection volume, used to calculate the concentration, gets smaller as the particles get larger. Further details concerning the data processing and quality control of the CIP images can be found in Crosier et al. (2011). Particles are separated into ice and liquid categories based on their circularity, C:

$$C = P^2/4\pi A,$$

where P is the measured particle perimeter and A is the measured particle area (a minimum area of 50 pixels – equivalent to a spherical particle with a diameter of 200µm - is used as smaller particles are not sufficiently resolved to discriminate between drops and crystals) . Following previous studies (Crosier et al., 2011; Taylor et al., 2015), particles with circularities between 0.9 and 1.2 are classified as circular and therefore liquid drops. For particles with values from 1.2 to 1.4 the decision on whether to count the particles as liquid or ice was made on a flight-by-flight case after looking at images. Particles with values over 1.4 were always counted as ice. The efficacy of the phase separation and choice of the circularity thresholds was confirmed by examining the sorted images 'by-eye'. The ice water content was calculated using the Brown and Francis mass-dimension parameterisation (Brown and Francis, 1995).

For this study it has been assumed that the clouds are mostly mixed phase clouds and that the particles observed by the CAS (less than 50 µm) are all liquid, while the CIP particles are characterised as either liquid drops or ice depending on the value of their circularity (see above). The number of large liquid drops (greater than 50 pixels - 200 µm equivalent diameter) seen by the CIP is very small in all flights. The total number of particles seen by the CIP is much lower than that observed by the CAS and the number of particles, seen by the CIP, less than 50 pixels (equivalent to a circular drop of 200µm diameter) are for all flights less than 2-3% of the number seen by the CAS and so we have ignored these particles whose phase we do not know. The version of the CAS probe used for this study was not able to measure polarization and so it was not possible to attempt to discriminate between solid and liquid with the CAS, however, it seems likely that the assumption that all CAS particles are liquid is a valid one particularly as we find the liquid water calculated assuming the CAS particles are all liquid agrees reasonably with that calculated from the hot wire probe on the CAPS (as stated above) Moreover we see a distinct peak (not shown) in the CAS size spectra, when in mixed phase clouds, indicative of drop formation.

In this study the average cloud properties over all the flights shown in Fig. 1 are considered. In-cloud data was averaged over one-degree longitude bands. In-cloud periods were defined as when the CAS number concentration for particles larger 4 µm is greater than 1 cm$^{-3}$ or when the CIP detects ice particles. The averaged values reported here for each longitude and each year are calculated by first averaging the values for each point for each flight and then averaging the individual flight averages over

the whole campaign. This means that each flight has the same weight giving a more representative final value. This averaging method also means that a long flight in a particular longitude bin does not dominate the overall average.

The CAS instrument has also been used to examine the aerosol concentrations outside clouds. To improve statistics all flights made in the study area during 2010 and 2011 when the CAPS probe was operational have been used. This includes an extra 31 flights that were primarily to investigate the boundary layer or the large scale flow but still had clouds present in the sky. To ensure that these measurements only include cloud free conditions the data was filtered by removing periods when there were particles larger than 1 μm. The CAS instrument only measures particle larger than 0.5 μm and so only gives us a measure of the larger aerosols but it is assumed that this will bear some relation to the number of CCN and INP available.In the case of INP the relationship between aerosols greater than 0.5μm and INP is represented in the parameterisations developed by (Demott et al., 2015 and DeMott et al., 2010)).

**2.2 Meteorological conditions**

Figure 3 shows the mean sea level pressure from the ERA interim reanalysis for the periods of the two campaigns. The flow during 2010 is generally slack while in 2011 Amundsen Sea low to the west of the Antarctic Peninsula has intensified and moved east. This has brought a more northerly flow across western side of the peninsula and this could be expected to bring warmer air. However, looking at the temperatures as a function of longitude from the aircraft flights in figure 4 we see that 2011 is actually colder in the west and this is also seen in the temperature fields from the ERA reanalysis as well as in the radiosonde ascents performed daily at Rothera Station (not shown). The cold in the west is a result of air being pulled around the tip of the peninsula from the Weddell Sea and this is confirmed by the back trajectory analysis reported later in this paper. The relative humidity (RH) plotted as a function of longitude (figure 5) shows more variability than the temperature record and there is no clear difference between the years, except an increase of RH in the West in 2010 in spite of the larger temperatures (figure 4), what indicates an increased amount of water vapour at that time

**3 Results on clouds and aerosol measurements.**

**3.1 Liquid phase in clouds**

The number concentration of liquid drops (cm$^{-3}$) averaged over one degree longitude bands is shown in Fig. 6. Each year is plotted separately with 2010 in black and 2011 in red. At each longitude the average from each individual flight is shown as a point – the small number of points at each longitude mean that it is not reasonable to calculate the standard deviation but the spread of the points gives some idea of the data variability. To improve statistics the data was binned further into two large bins on each side of the Peninsula, one from 67 to 74˚W and the other from 60 to 65˚W. These values, along with their standard deviations, are reported in Table 1. The CAS size spectrum showed peaked distributions around 8-12μm illustrating the condensational growth of supercooled droplets (not shown). The average number of droplets varies from around 60 to over

200 cm$^{-3}$ (Fig. 6), which is typical of concentrations found over the open ocean away from possible sources of CCN from continental land masses (Pruppacher and Klett, 1997; Chubb et al., 2015). Table 2 gives the statistical significance of differences between both sides of the Peninsula as well as differences between both years, using a t-test.  The most significant difference (at the 99% level) for liquid drops is on the east of the Peninsula between 2010 and 2011. A less significant but still

noticeable difference exists between either sides of the Peninsula in 2011 (90%).These relationships were not found in 2010 (<80%).

The liquid water content (LWC) in clouds averaged by longitude for the two years is shown in Fig. 7. Again to get better statistics the values of LWC have been averaged on both sides of the Peninsula using the same size longitude bins that were used for liquid drop numbers. A significant difference between the two years is found at the 90% level in the east and 95%

level in the west (see Table 2), although the east-west difference on both years is not significant. The single point of high LWC in 2011 at 73°W is the result of a small number of flights into active frontal systems with large amounts of liquid water.

### 3.2 Ice phase in clouds

The ice particle numbers and ice water content are shown in Fig. 8 and 9 for areas in the cloud that were at least partially glaciated – that is non circular particles were observed in the CIP. The number of ice particles observed is roughly 5 to 6 orders

of magnitude lower than the number of cloud droplets and the relative amplitude of the variability higher (the standard deviation of liquid droplets number concentration is about 50% of the average values, while it is as high as or higher than averages for ice crystals number concentrations).  Also not all the clouds investigated were glaciated to any extent and so there are slightly fewer measurements (table 1) for ice crystals than for the drops – except for the east of the Peninsula in 2010 when there was an observation of a completely glaciated cloud. First looking at the number concentration of crystals, Fig. 8 and

Table 1 show that in 2011 there were more crystals on both sides of the Peninsula than in 2010. However, Table 1 shows the standard deviation of the crystal numbers is large. Table 2 shows that differences are not significant between the two years (<80%) on the east, but significant on the west (95%). Differences are also significant between either side of the Peninsula in 2011 (at the 95% level), however not in 2010 (75%)

The ice water content (Fig. 9) shows a similar trend to crystal numbers in Fig. 8. However, in this case using the averaged

values on each side of the Peninsula (Table 1) the significance is higher, the difference between both sides in 2011 and between both years on the west are significant at the 95%level.

The distribution of crystals with atmospheric temperature for the two years is shown in Fig. 10. Median ice crystal number concentrations have been derived over 0.5 °C bins along with the associated median absolute deviations. The distributions are very different in the two years and between the east and west of the Peninsula. In the west, both years have a peak ice number

concentrations around -5°C. As reported by Grosvenor et al. (2012), for some of the 2010 flights only, this peak is most probably related to a secondary ice production process known as the Hallett-Mossop process (Hallett and Mossop, 1974). This

consists of droplets shattering upon freezing when impinged by existing crystals (riming process), thus causing a cascade of crystals. This process only operates efficiently in narrow temperature range (approximately -3° to -8°C) with an optimum around -5°C, and requires large enough ratio of number concentrations nS of small drops (< about 13 µm) to large drops nL (> about 24 µm) (Mossop, 1985). Typical observed concentrations by Mossop (1985) of large (small) drops were 10-80 cm3

(5-60 cm3) with the ratio nS / nL ranging between 0.1 and 5. In the present work, median values per flights for nS, nL, and nS / nL (where peaked concentrations are observed at -5°C) are in the following respective ranges, 16-90 cm-3, 9-33 cm-3, and 0.5-13 (with flight minima always above 0.1). Note that no constrain is available in the present dataset to properly estimate the rimers velocity. However, Mossop (1985) suggests that splintering can occur at rimer velocity as low as 0.2 m s-1, and observes no abrupt drop of splinter production from 1 to 0.55 m s-1 , meaning that the rimer velocity does not show a lower cut-off

preventing Hallett-Mossop from happening. In addition to that, other mechanisms known to trigger ice multiplication do not operate in the -8°C to -3°C temperature range but at colder temperatures (<-10°C), for example ice-ice collision (Takahashi et al., 1995), (Yano and Phillips, 2011), freezing of large drops with ice spicules formation (Lawson et al., 2015), or breakup of crystals, which preferably requires irregular shapes like dendrites that are favoured at low temperatures (Bacon et al., 1998). The main limitation to an absolute identification of the process as Hallet-Mossop is the resolution (both temporal and spatial)

of the CAPS probe to observe the process in detail from its start as only the resulting larger crystals are observed and not the first ice.

In the east there is no clear peak at the high temperatures (>-10 ˚C) in 2011; there is only a small peak in 2010. It suggests that relatively less secondary ice production is observed for the eastern side of the Peninsula. It can be seen that during 2011 the number of ice crystals increase from -10°C to around -20˚C and this probably corresponds to primary ice production (where

an INP active at a given temperature interacts with a droplet or vapour to form one crystal).  Ice Nuclei parameterisations relying either only on the temperature (Cooper, 1986) or both on the temperature and the aerosol (>0.5µm) concentration (DeMott et al., 2010) are plotted in Fig. 10. They are meant to account for primary ice nucleation, and predict increasing INP concentrations from -10°C to -20˚C. Note that for the DeMott's parameterisation aerosol concentrations bracketing the present average observations (section 3.3, Figure 11) were used. Comparing the trends of either of the parameterisations to the ice

crystal distributions also strengthen the idea that the ice crystal distributions peaking around -5°C are to be related to secondary ice production processes.

## 3.3 Aerosols out of clouds

It is to be expected that the number of drops and ice crystals will be controlled by the number of aerosols acting as CCN and

30 IN, excluding for the moment the role of secondary ice production.  The instruments that were fitted to the Twin Otter in 2010 and 2011 did not allow the full range of aerosols to be measured. However the Cloud and Aerosol Spectrometer (CAS), that

is part of the CAPS probe, measures aerosol size distribution down to 0.5µm. The zonal variation of aerosol particles greater than 0.5 µm and less than 1µm is shown in Fig. 11, and their average numbers in both years and both sides of the Peninsula are given in Table 1. Note that to have a better statistics and a better picture of the aerosol population in the region 55 flights were used, ie including flights not primarily intended to cloud measurements, but that were still equipped with the CAPS

probe. Table 2 gives the significance of aerosol differences between years and regions. If the single point in the far west in 2011 is excluded (which is from one flight and is considered an anomaly) the two years are very similar to the west of the Peninsula.  To the east there is a significant difference between the two years (at the 95% level) with 2011 having almost twice the concentration of aerosol as 2010. In 2011 there is also a significant difference between the east and west sides of the Peninsula (at the 99% level) with twice the concentration of aerosol on the east as on the west of the Peninsula. Importantly,

the same conclusions are drawn when using the 24 cloud-flights only.

**4 Discussion**

The variability of cloud droplet and ice concentration observed in this study is quite large and the number of flights made is small, compared to the number of measurements that have been made at mid-latitudes. This makes identifying statistically-significant changes between geographical areas or between years difficult.  This problem has been dealt with by first averaging

the data into longitudinal bands (Figs 6,7,8,9 and 11). Although this allows some of the differences of interest to be observed, there are still too few points to determine if these differences are significant. To help with these, averages were taken of all the data on each side of the Peninsula (Table 1) and in this case clear statistical differences can be seen (Table 2).

**4.1 Aerosol source regions and liquid droplets**

The average number of clouds droplets in the clouds (see Fig. 6) varies over a large range from just a few to almost 300 cm$^{-3}$

and this probably reflects the large range in CCN found in the coastal areas of Antarctica. The mean number of drops on each side of the Peninsula are not untypical of values to be found in mid ocean (Pruppacher and Klett, 1997; Chubb et al., 2015) and reflect the position of Rothera exposed to air masses with origins in the Southern Pacific. Table 2 shows that it is only in the east that there is a significant difference (at the 99% level) between the two years in the aerosol concentration and this might be a result of the different source regions for particles between the two years.

To test this hypothesis back trajectory analysis using the Hybrid Single-Particle Lagrangian Integrated Trajectory (HYS- PLIT) model has been performed (Stein et al., 2015). Seven day back trajectories were calculated using the National Centers for Environmental Prediction (NCEP) reanalysis meteorological field with starting points located at 60 s intervals along the track of each flight. Figure 12 shows the position of the low altitude ($\leq$ 300 masl, see below) air masses 48 hours backwards, along with their altitude (colour coded), for both years and sides of the Peninsula. The sea ice fractional coverage (at 25 km

resolution), obtained from Nimbus-7 SMMR and DMSP SSM/I-SSMIS Passive Microwave Data (Cavalieri et al., 1996), has

been over plotted in blue. As in section 3.3 (and in Figure 11) 55 flights were used to investigate the average origin of the air masses on both years. A striking difference appears between 2010 and 2011, with 2011 showing more sources above the sea ice. Table 3 summarizes the relative proportion of air masses passing above sea ice compared to above open water/ice shelf (referred to as "other") among all low-altitude air masses (i.e. below 300 m, approximate boundary layer height (Fiedler et al.,

2010)), and along all the back trajectories. The relative proportions are also indicated for the case when only the 24 clouds flights are used (number in parentheses in Table 3). We first comment on the analysis using the 55 flights. A sea ice covered region was defined as a region where fractional coverage is larger than 1% (however the sea ice regions of interest in table 3 normally have much larger values, see Fig. 12). These low altitude air masses will have greater sensitivity to surface aerosol emissions. The statistics integrated over different durations are shown to illustrate the consistently dominating or increasing

trends of the relative contributions of sea-ice low altitude air masses among all low altitude air masses. Whatever the time interval, low altitude air masses sampled to the east of the Peninsula always overpass sea ice regions relatively more often than they do overpass other regions – with the influence of the sea ice being greater in 2011.  Looking at table 3 we can see that the sea ice has most influence, over most time periods, in the east in 2011 then less in the east in 2010, less again in west in 2011 and least in the west in 2010. Thus, Table 3, along with Fig. 12 shows that (a) air sampled on the east is more sea-ice influenced

than air sampled on the west in both years, (b) air sampled on the east in 2011 has had a relatively longer sea-ice track at low altitudes than in 2010 and (c) air sampled on the west in 2011 is slightly more sea-ice influenced than in 2010. Looking at the analysis restrained over the 24 cloud flights only, the above (b) statement becomes less clear as only the analysis over the period covering the previous 48 hours shows a clear increase of sea ice influence on the east in 2011 (Table 3, second line), compared to 2010, but statement (a) and (c) still prevail and are even strengthen when considering this smaller statistics.

Overall clear differences appear in the origin of the air masses when splitting the analysis into west/east and 2010/2011 comparisons.

The larger concentration of aerosols in 2011, especially on the eastern side (Fig. 11), compared to 2010 (Table 2), could be explained by more air masses having longer tracks at low level over sea ice covered regions of the Weddell Sea. It is possible that sea salt on the snow covered surface of the sea ice could easily be lofted into the air as blowing snow which sublimes to

form sea salt aerosol, what has been suggested as an efficient mechanism for getting sea salt aerosol (Yang et al 2008), which could act as CCN, into the boundary layer. This suggests that the increase in droplet number concentrations seen in 2011 on the eastern side of the Peninsula (Fig. 6) could result from the increase in aerosol numbers. Fig. 13 shows average vertical profiles of aerosol number concentration on both sides of the Peninsula for 2010 (blue circle) and 2011 (red triangles),for the 55 flights used in section 3.3 (Note that the same vertical profiles plotted with the 24 cloud-flights only display quantitatively

similar differences). The corresponding shaded areas indicate absolute minimum and maximum (non-null) values at each altitude level over the respective campaigns. Like the latitudinal averages of aerosols shown in Fig. 11, Fig. 13 shows that the interannual differences in aerosol concentration on the eastern side is much more pronounced (see the 99% significance level in Table 2) than on the western side (93% significance level). Interestingly, these differences occur almost exclusively below

2000-2500 m. Above that altitude, the concentration of aerosols is similar between both years and both sides (around 0.1-0.2 cm$^{-3}$). The central part of the Peninsula between 67°W and 65°W is also characterized by this concentration of about 0.1 cm$^{-3}$ (not shown). This suggests that a homogeneous mixture of aerosols (at least in terms of number concentration, if not nature) prevails at altitudes above the height of the Peninsula mountain barrier  Conversely, at lower levels, the Antarctic Peninsula

barrier would help sustain different pools of aerosols on either of its sides. These aerosol pools could give rise to different cloud microphysics on either side of the Peninsula, and a possible interpretation of the observations is that the increased number of droplets east in 2011 compared to 2010 could be the result of an environment more influenced by particle originating from sea ice regions.

### 4.2 Ice production, altitude ranges, and aerosols

Interestingly, the ice crystals number concentration (Fig. 8) does not follow the same pattern as the liquid droplets. A significant difference (at the 95% level) is seen between the two years in the west and also a similar significant difference is seen between both sides of the Peninsula in 2011. However, this is not unexpected, as while aerosols are the source of CCN and INP so that a correlation between the numbers of liquid drops and ice crystals might be expected, there is another process going on to create ice crystals. This is the secondary ice production due to the Hallett-Mossop process for temperatures warmer than -10°C

(introduced in section 3.2). The peak of ice crystal numbers at temperatures above -10°C can be clearly seen (Fig. 10), especially in the west where as might be expected the temperatures are warmer. This is illustrated by Figure 4 which shows each year's temperatures averaged by latitude along the aircraft tracks. In the east there is much less evidence of secondary ice formation for both years (Fig. 10) and this correlates with the similarly lower temperatures in both years (Fig. 4). At temperatures below -10°C (Fig. 10) it can be seen that in 2011 there is a distinct peak in ice concentration, which suggests

more primary ice production (see section 3.2) in both the west and the east at temperatures approximately ranging between -10 °C and -20°C, while there is almost virtually no primary ice production peak in 2010. Fig. 14 shows the entire dataset of crystal distribution with temperatures for the two years (both sides at the same time) colour coded with the altitude of the observation. The primary ice production mainly occurs at altitudes above roughly 2500 meters, while the secondary ice production occurs almost exclusively below 2000 meters – this of course is related to atmospheric lapse rate. This means that

the primary ice production peak presence cannot be solely related to the increase of aerosols, which is observed mostly below 2500 meters (see section 4.1). As such, the lower primary ice production in 2010, and its presence in 2011 on both sides of the Peninsula can be linked to the colder 2011 temperatures compared to the 2010 above 2500 m (not shown). Another possible factor in the lower ice production in 2010 is the different nature of INP related to the different air masses which would feed with aerosols both sides of the Peninsula. It can be seen that in 2011 more air masses were coming from above sea ice regions

(see section 4.1). It has been suggested that biogenic particles found in sea ice could act as INP (Burrows et al., 2013) and the same method for transferring from sea ice to the atmosphere (as the sea salt) could explain the different nature of INP found in 2011 above 2500 m, leading to more primary ice production in 2011.

## 5 Summary and conclusion

Fig. 15 summarises the results reported in this paper. Our observations show the significant differences in cloud properties between the two measurement periods, February 2010 and January 2011, and between measurements made to the east and to the west of the Antarctic Peninsula, and present some possible explanation for those.

5   January 2011 showed almost twice as many aerosols on the eastern side of the Peninsula than on the western side, and more than in the previous year. The larger number of droplets in 2011 can be explained by an increase of CCN that can be inferred from the observed increase in large aerosols (>0.5 µm) in that year. The larger number concentrations of aerosols can be linked to the different source regions. In 2011, relatively more air masses were coming from the sea ice, and relatively more on the eastern side than on the western side. This bringsa possible interesting explanation to the twice larger number of droplets in 10   2011 east (approx. 200 cm$^{-3}$) compared to 2010 east, and also 50% larger than 2011 west.

The Hallett-Mossop secondary ice multiplication (at temperatures warmer than -10°C, peaking around -5°C) process seems to be key to ice production mechanism below 2500 meters on both sides of the Peninsula, and mainly on the warmer western side. In contrast above 2500m primary ice production mechanism is expected to dominate. The larger production of primary ice crystals (at temperatures colder than -10°C, peaking around -20°C) above 2500 meters in 2011 compared to 2010 on both 15   sides of the Peninsula, is due to colder temperatures (activating more IN) and the possible different nature of INP (coming relatively more from above sea ice regions). Indeed, there is only a small increase in number concentration of large (>0.5 µm) aerosols in 2011 above 2500m. This is why the increase in aerosol number supports the larger amounts of liquid droplets, but does not seemingly support the larger number of primary ice seen above 2500m in 2011 on both sides of the Peninsula.

These observations show that the concentration of large aerosols (>0.5 µm) is fairly similar across the Peninsula (about 0.1-20   0.2 cm-3) at altitudes higher than the mountain barrier where dynamics would sustain a well-mixed aerosol population. Conversely, the mountains would favour the creation of so called aerosol pools on either side of the Peninsula, with different concentrations and nature that would be responsible of a different microphysics. Similarly, ice production is affected by the temperatures prevailing on either side of the Peninsula, with little secondary ice production occurring on the eastern (colder) side, and more on the (warmer) western side. The mountains height controls the altitude at which both sides display similar 25   primary ice production peaks above 2500m, because of similar population of aerosols.

The number of liquid drops and primary ice crystals is correlated with the sources of the air masses. These results indicate that the sea ice covered Weddell Sea could be a more important source of CCN and INP than the open ocean. This may have more

general implications for the microphysics of clouds that cover the Southern Ocean. The Southern Ocean is an area in which large errors have been identified in the simulated cloud cover, leading to large radiation biases in global climate models (Flato et al., 2013; Bodas-Salcedo et al., 2014). Given that in winter, sea ice can extend up to 60˚S and even 55˚S in some regions, sources of CCN and INP related to the sea ice could potentially have large impact on the microphysics of clouds forming over the Southern Ocean, as they seem to have across the Antarctic Peninsula.

The present study is the very first of its kind attempting to depict cloud microphysics and aerosols across the Antarctic Peninsula from a small amount of flights, the scenario we suggest hopefully will stimulate other studies and measurements to better assess the plausibility of our interpretations.

*Acknowledgements*. This work would not have been possible without the help of the scientists and support staff who helped in the Antarctic. The work was funded by UK Natural Environment Research Council with core funding and under grant NE/K01305X/1.

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

**Table 1: Average values for cloud measurements and out-of-clouds aerosols for both years and both sides of the Peninsula. $\sigma(x)$ refers to the standard deviation of the variable x, and N the number of flight averages available for the global Eastern or Western averages. LWC refers to Liquid Water Content and IWC to Ice Water Content.**

|  | West 2010 | West 2011 | East 2010 | East 2011 |
|---|---|---|---|---|
| Drops (cm$^{-3}$) | 124.27 | 154.62 | 103.26 | 191.68 |
| $\sigma$(Drops) | 68.6 | 63.52 | 68.98 | 71.62 |
| N | 23 | 23 | 20 | 22 |
| LWC (g kg$^{-1}$) | 0.094 | 0.157 | 0.069 | 0.108 |
| $\sigma$(LWC) | 0.07 | 0.12 | 0.06 | 0.07 |
| N | 23 | 23 | 20 | 22 |
| Ice crystals (L$^{-1}$) | 0.388 | 1,414 | 0.244 | 0.410 |
| $\sigma$(Ice crystals) | 0.362 | 2,038 | 0.392 | 0.428 |
| N | 18 | 21 | 21 | 20 |
| IWC (g kg$^{-1}$) | 0.00314 | 0.00842 | 0.00177 | 0.00332 |
| $\sigma$(IWC) | 0.00308 | 0.00848 | 0.001624 | 0.00368 |
| N | 18 | 21 | 21 | 20 |
| Aerosols (cm$^{-3}$) | 0.0692 | 0.105 | 0.106 | 0.218 |
| $\sigma$(aerosols) | 0.0678 | 0.129 | 0.153 | 0.314 |
| N | 53 | 77 | 59 | 87 |

**Table 2: Statistical significance of the differences between either year on either side of the Peninsula as obtained from the t-test performed for the four cloud variables and the aerosols (See text for details). Values greater than 90% are highlighted in bold.**

|  | N(drops) | LWC | N(ice) | IWC | N(aerosols) |
|---|---|---|---|---|---|
| W2010-W2011 | 83% | **96%** | **96%** | **98%** | **93%** |
| E2010-E2011 | **99%** | **94%** | 79% | **91%** | **98%** |
| E2010-W2010 | 68% | 78% | 75% | **92%** | 88% |
| E2011-W2011 | 93% | 89% | **96%** | **98%** | **99%** |

**Table 3: Relative proportion of low altitude (<300 meters) air masses passing over sea ice covered regions with respect to the total number of low altitude air masses along all back trajectories derived from HYSPLITmodel for 55 flight tracks from both campaigns (See text for details). In parentheses, the same percentages but for the 24 flights only Relative proportions are indicated for both years and both sides of the Peninsula, the side referring to the (start) ending point of the airmasses (back) trajectory. Percentages are computed over different time ranges, prior to reaching a given point of a flighttrack on either side. A region is considered as covered by sea ice as long as the sea ice concentration from the NIMBUS-7 MMR is larger than 1% (See section 4.1 for reference).**

|               | East 2011   | East 2010   | West 2011   | West 2010  |
|---------------|-------------|-------------|-------------|------------|
| Last 72 hours | 61%  (43)   | 35% (47)    | 37% (21)    | 24%  (9)   |
| Last 48 hours | 58%   (80)  | 45%  (60)   | 37% (22)    | 25%  (8)   |
| Last 24 hours | 48%   (94)  | 68% (91)    | 34% (26)    | 26%  (8)   |
| Last 12 hours | 97%   (100) | 90% (98)    | 33% (19)    | 15%  (7)   |

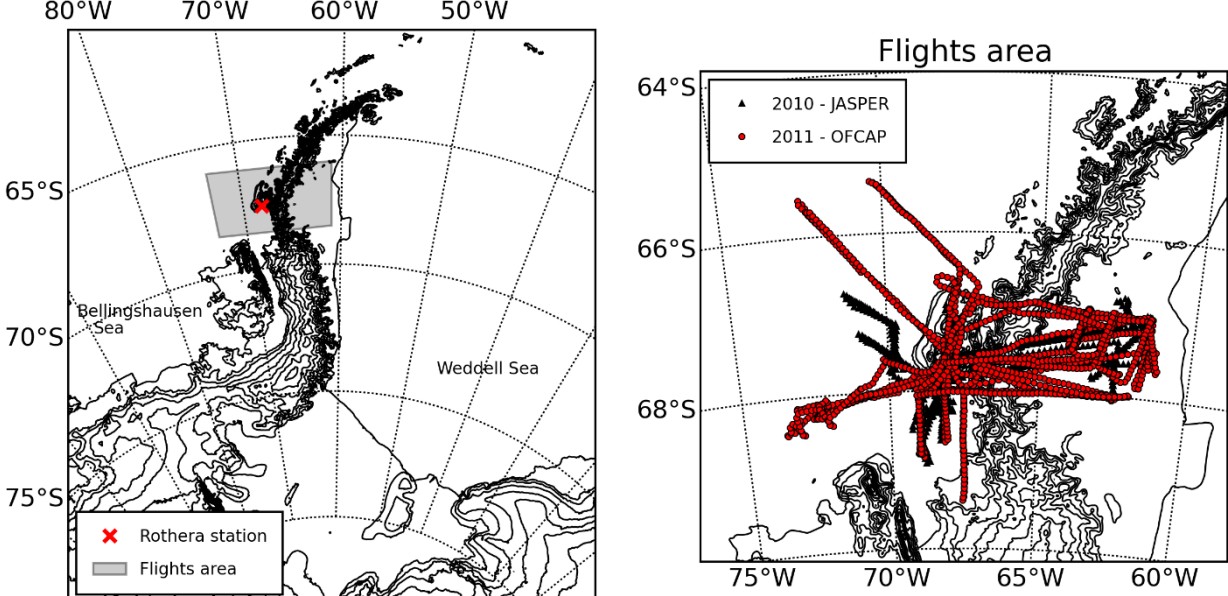

**Figure 1: (Left) The Antarctic Peninsula and flights area in context, with BAS Rothera station indicated (red cross). On the Peninsula's eastern side is the ice covered Weddell Sea. Solid lines indicate topographical features as well as ice shelves boundaries. (Right) Close-up on the flights area showing the flight tracks of both campaigns of interest with black triangles (February 2010) and red circles (January 2011). Topography derived by Fretwell et al. (2013)**

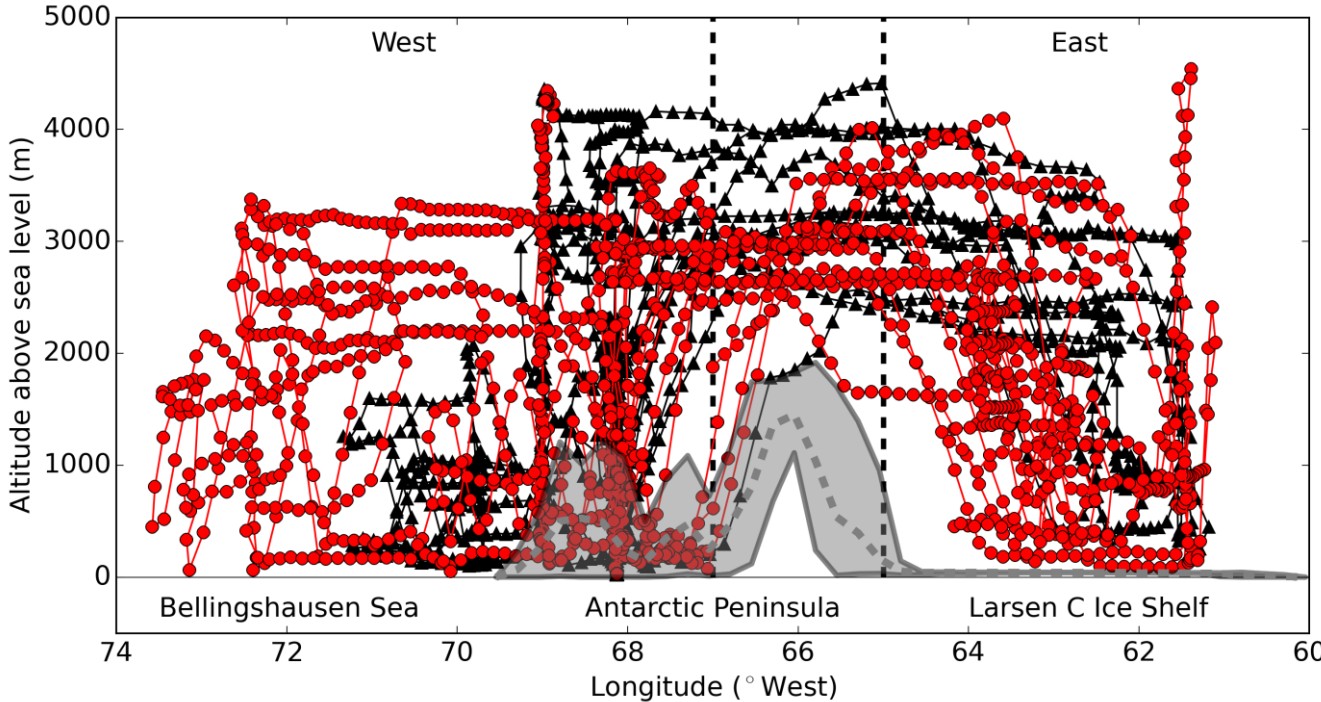

**Figure 2: Latitudinal cross section of flight tracks showing altitudes probed on both sides of the Peninsula. February 2010 (black triangles) and January 2011 (red circles). The grey shaded area delimits the maximum and minimum topography height (at 5 km resolution) between 68S and 67S and the grey dashed line shows the average topography height across the latitudes 68S to 67S. The vertical dashed lines delimits the Western (74-67 West) and Eastern (65-60 West) regions of the Peninsula as defined in this study.**

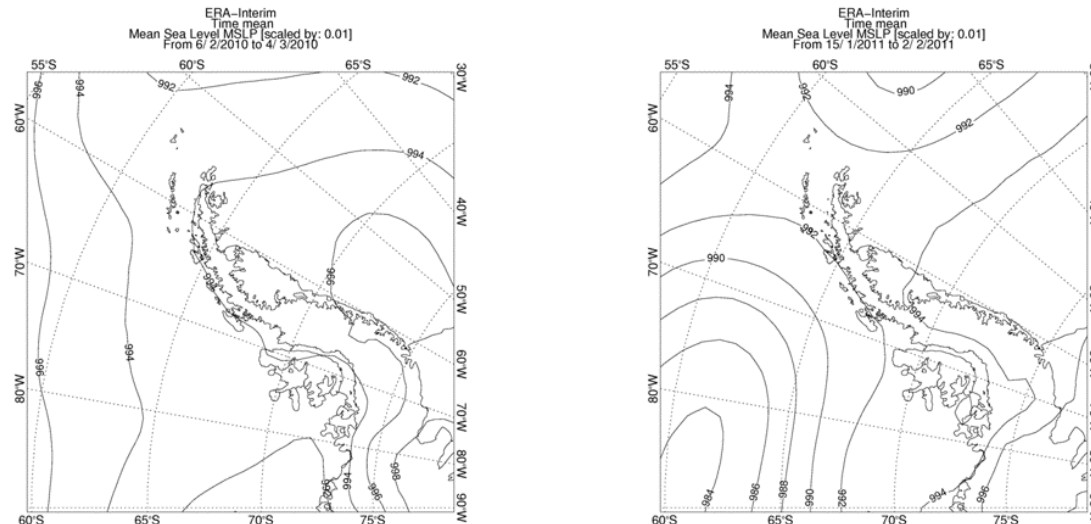

Figure 3: Mean sea level pressure from the ERA interim reanalysis for the periods of the aircraft campaign in 2010 (left) and 2011 (right)

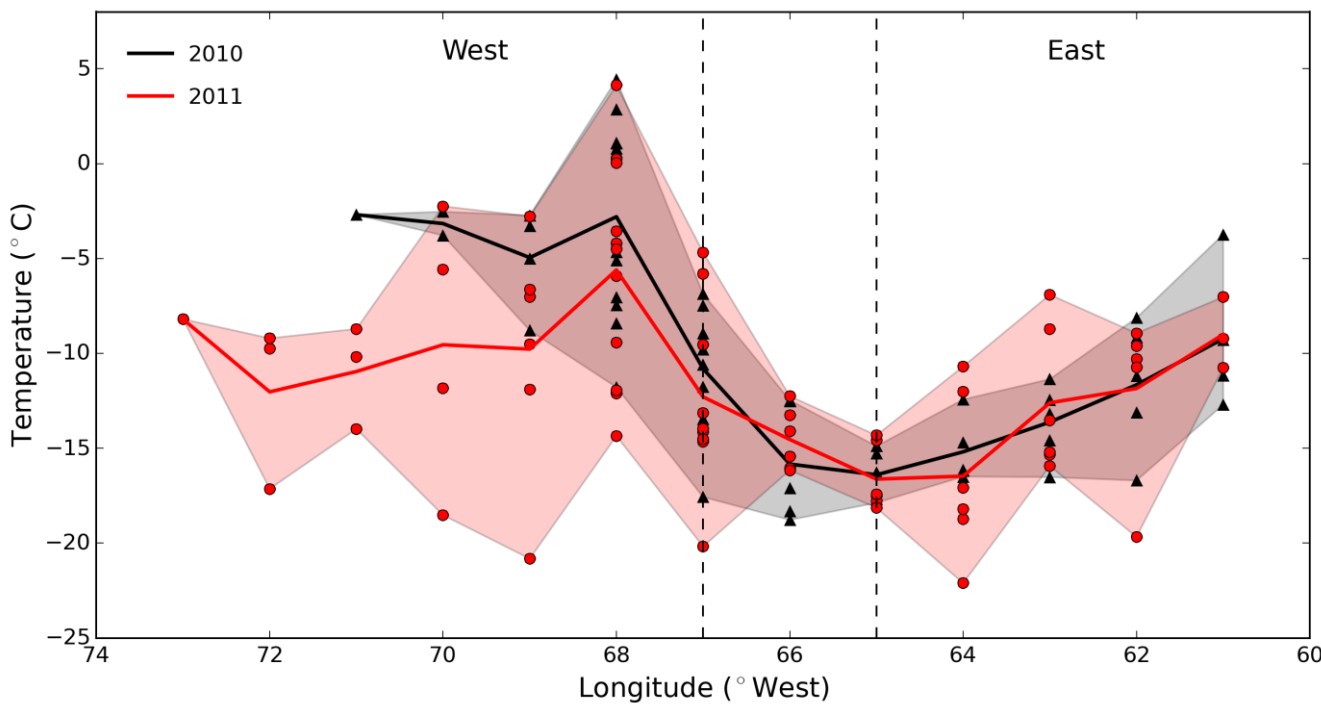

Figure 4: Atmospheric temperature as measured by the aircraft (°C) ) as a function of longitude, averaged in 1° longitude
bins. The flight averaged values are overplotted as black triangles (2010) and red circles (2011). Solid lines represent the
average of these averages in each longitude bin. Shaded grey (2010) and red (2011) areas indicate the spread of these
averages across the Peninsula. The vertical dashed lines delimits the Western and Eastern regions of the Peninsula as defined
in this study

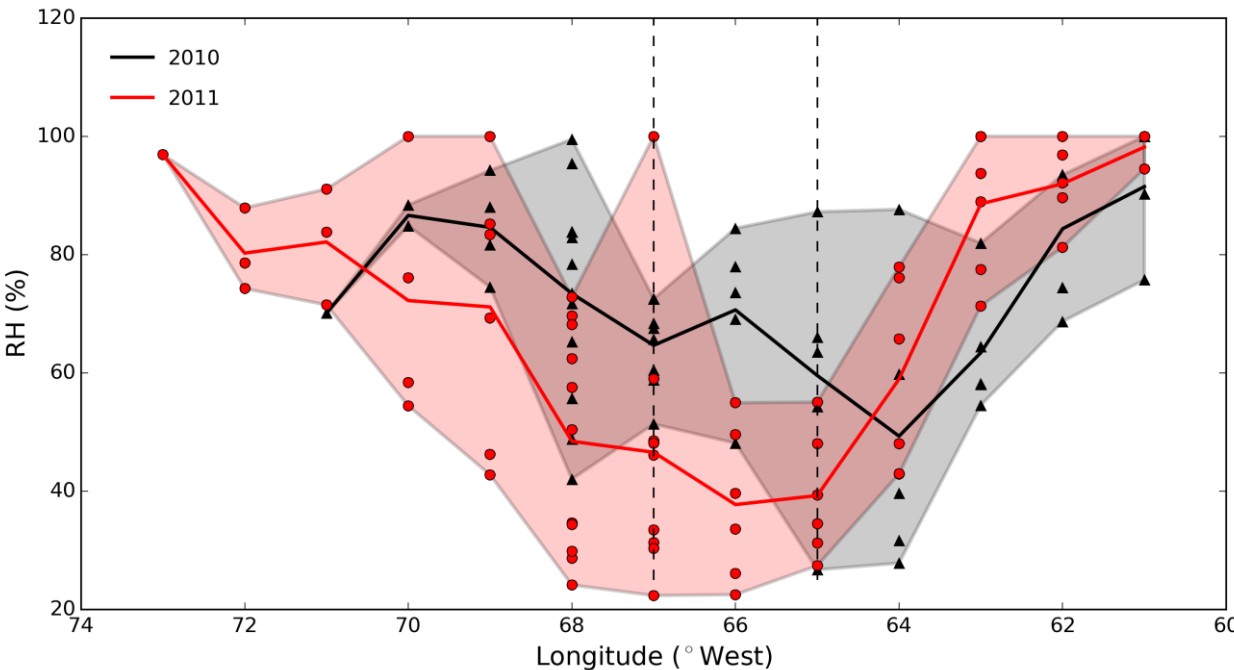

Figure 5: As figure 4, for the relative humidity,

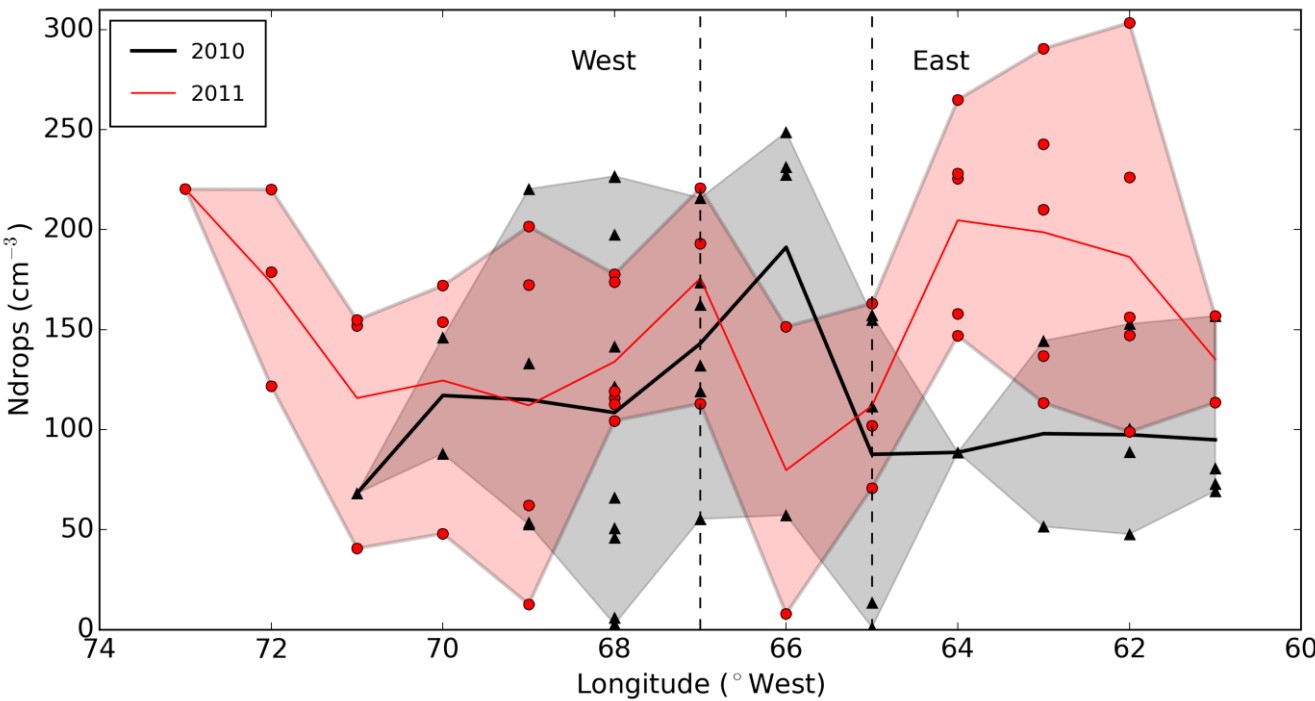

**Figure 6: Same as figure 4, for the Liquid droplets number concentration (cm⁻³)**

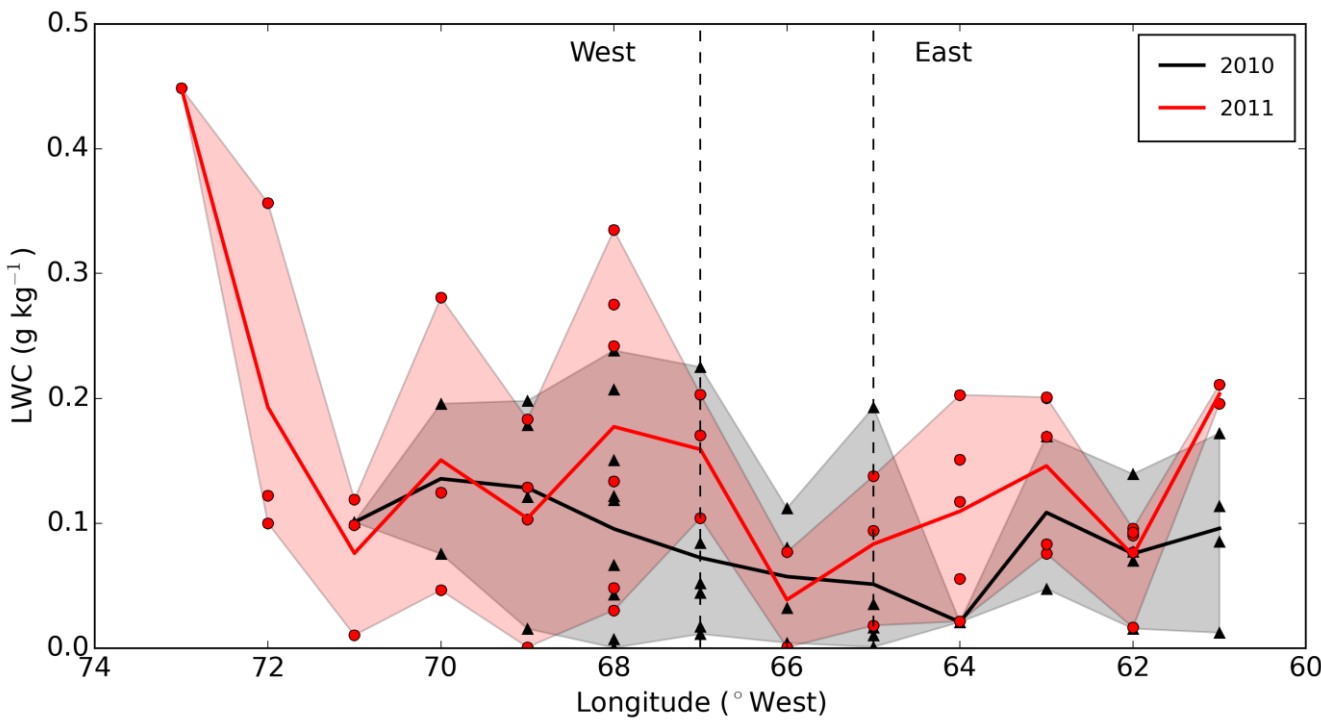

**Figure 7: Same as Figure 4, for the Liquid Water Content (LWC, g kg⁻¹).**

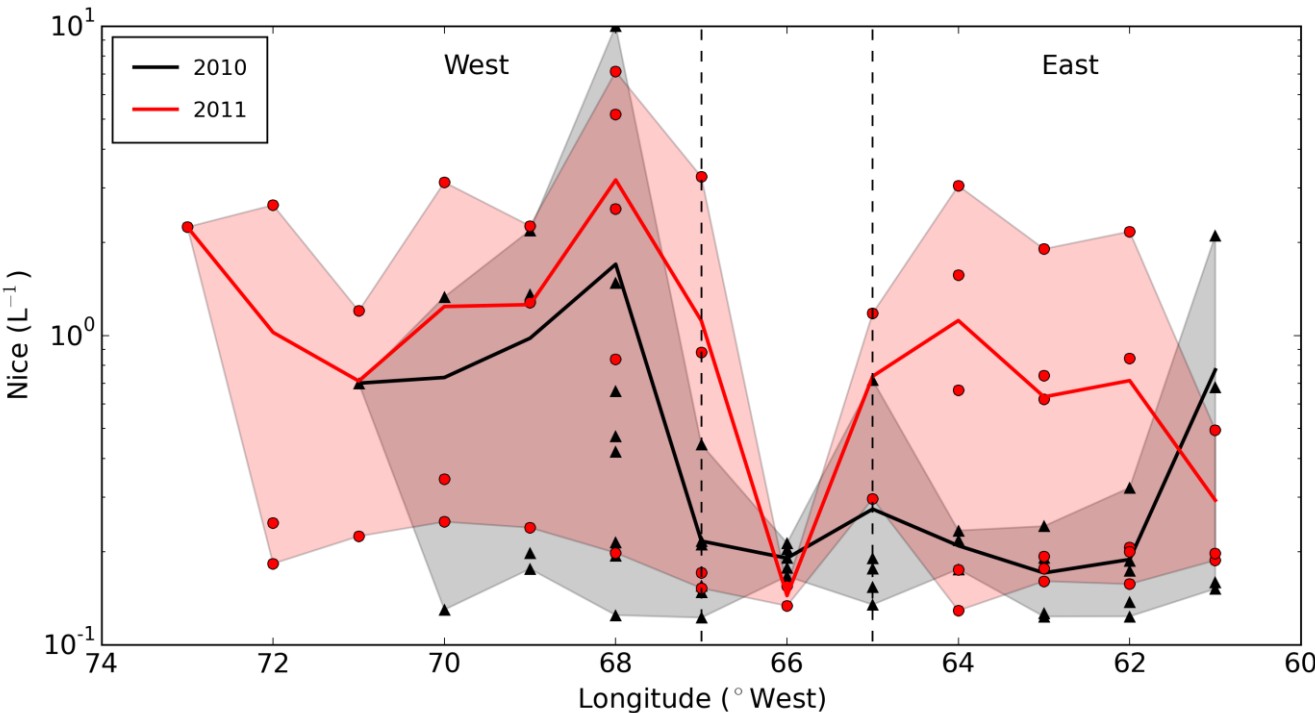

**Figure 8: Same as Figure 4, for the number concentration of ice crystals (L⁻¹).**

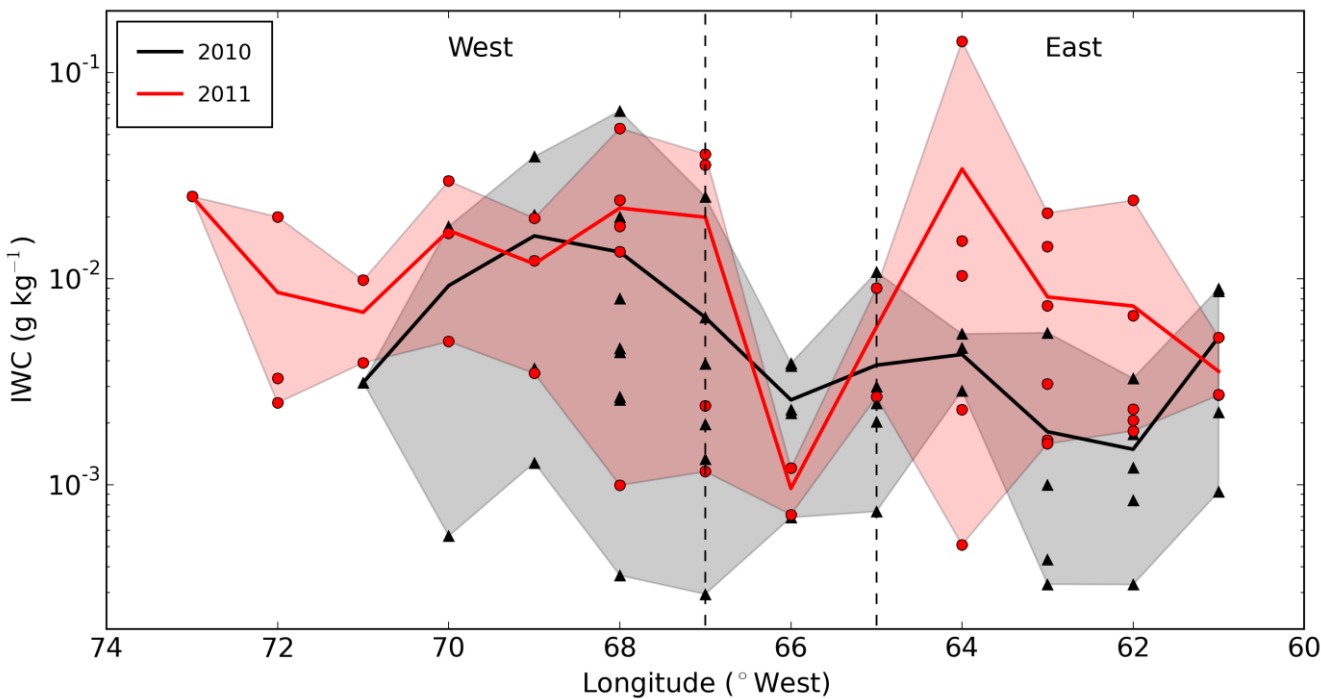

**Figure 9: Same as Figure 4, for the Ice Water Content (IWC, g kg⁻¹).**

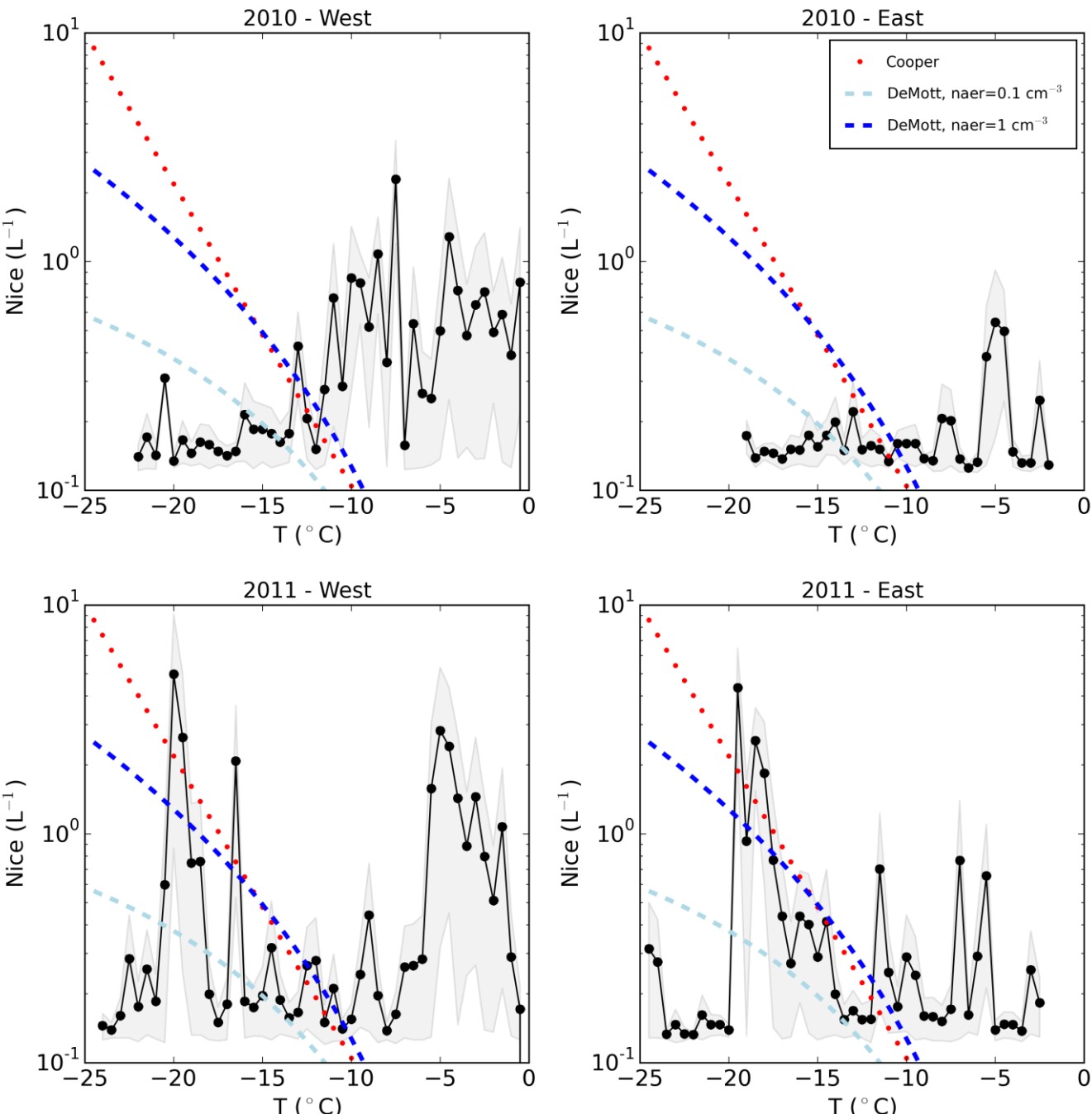

**Figure 10: Distribution of ice crystals (L⁻¹) as a function of atmospheric temperatures for 2010 (top row) and 2011 (bottom row), for the western side (left column) and for eastern side (right column). The median of the number concentrations was derived per 0.5°C bins. The dotted and dashed lines show the predicted number of primary ice particles from two parameterisation schemes (see text for details). The shaded area represents the median absolute deviation from the median value. West refers to 74-67°W and East refers to 65-60°W.**

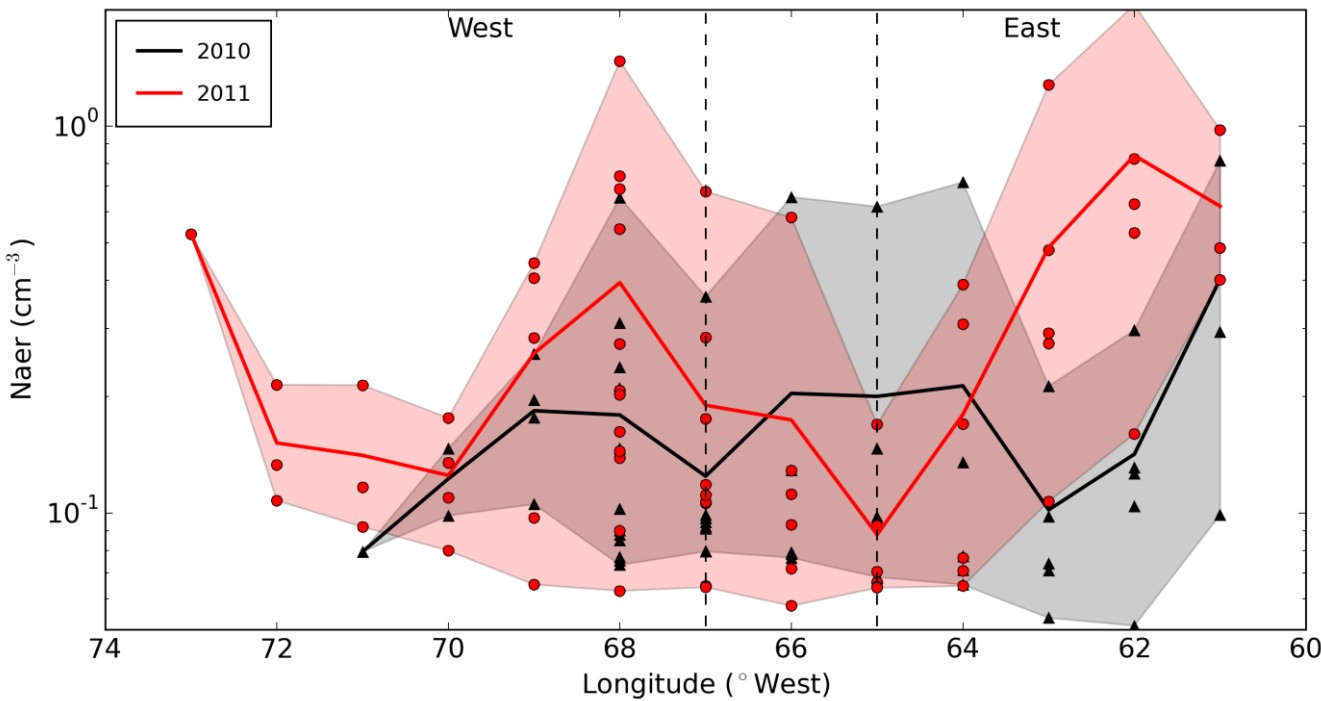

Figure 11: Same as Figure 4, for the number concentration of aerosols larger than 0.5 µm and smaller than 1 µm in diameter (cm⁻³).

55 flights are considered here (31 flights from both campaigns equipped with the CAPS probe but not intended to cloud measurements in addition to the 24 cloud flights) to increase the statistics and give better overview of the aerosol population across the peninsula for both years. See text (section 3.3) for more details. The vertical dashed lines delimits the Western and Eastern regions of the Peninsula as defined in this study

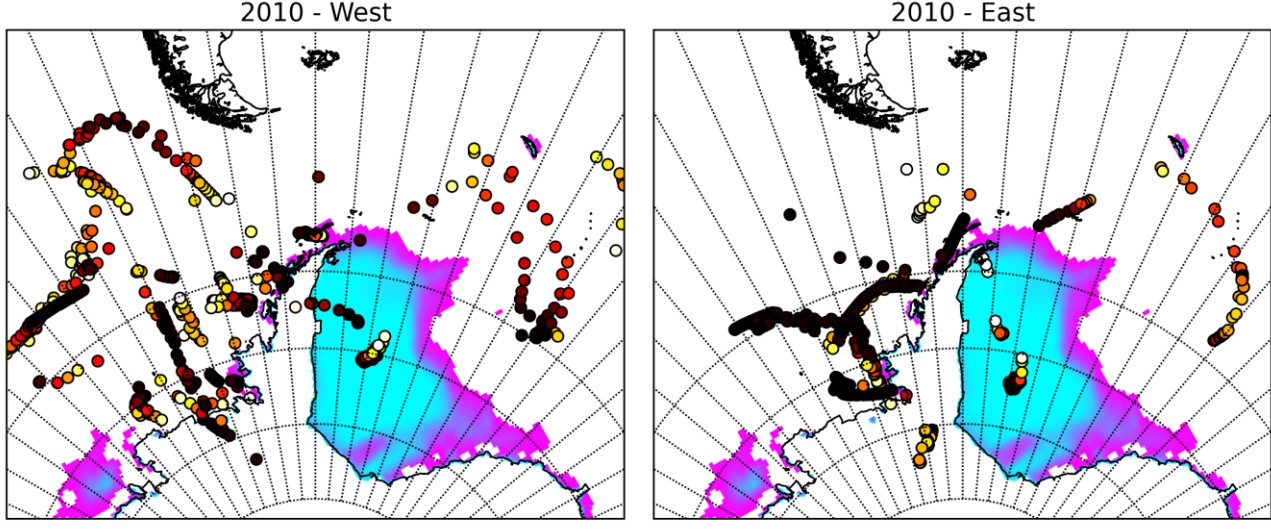

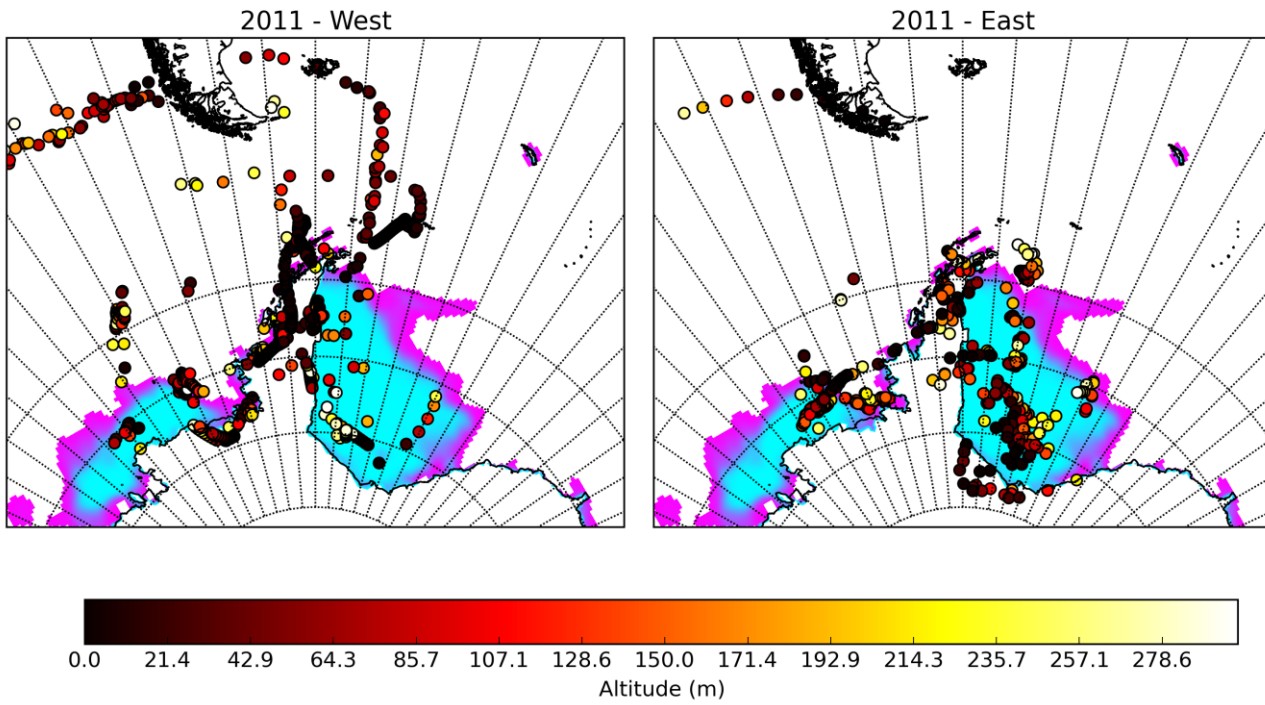

**Figure 12: Location, and altitudes (colour-coded, in meters), of low altitude (≤ 300 m – above ground) air masses 48 hours prior to ending on a flight track (from either of the 55 flights used in Figure 11) west (left column) or east (right column) of the Peninsula. Over plotted is the monthly averaged sea ice fractional coverage as obtained from Nimbus 7 – SMMR daily data at 25 km resolution (See text for reference) (lightest blue is >90%, darkest blue is 50%, dark magenta 25% and lightest magenta is below 1%). West refers to 74-67°W and East refers to 65-60°W.**

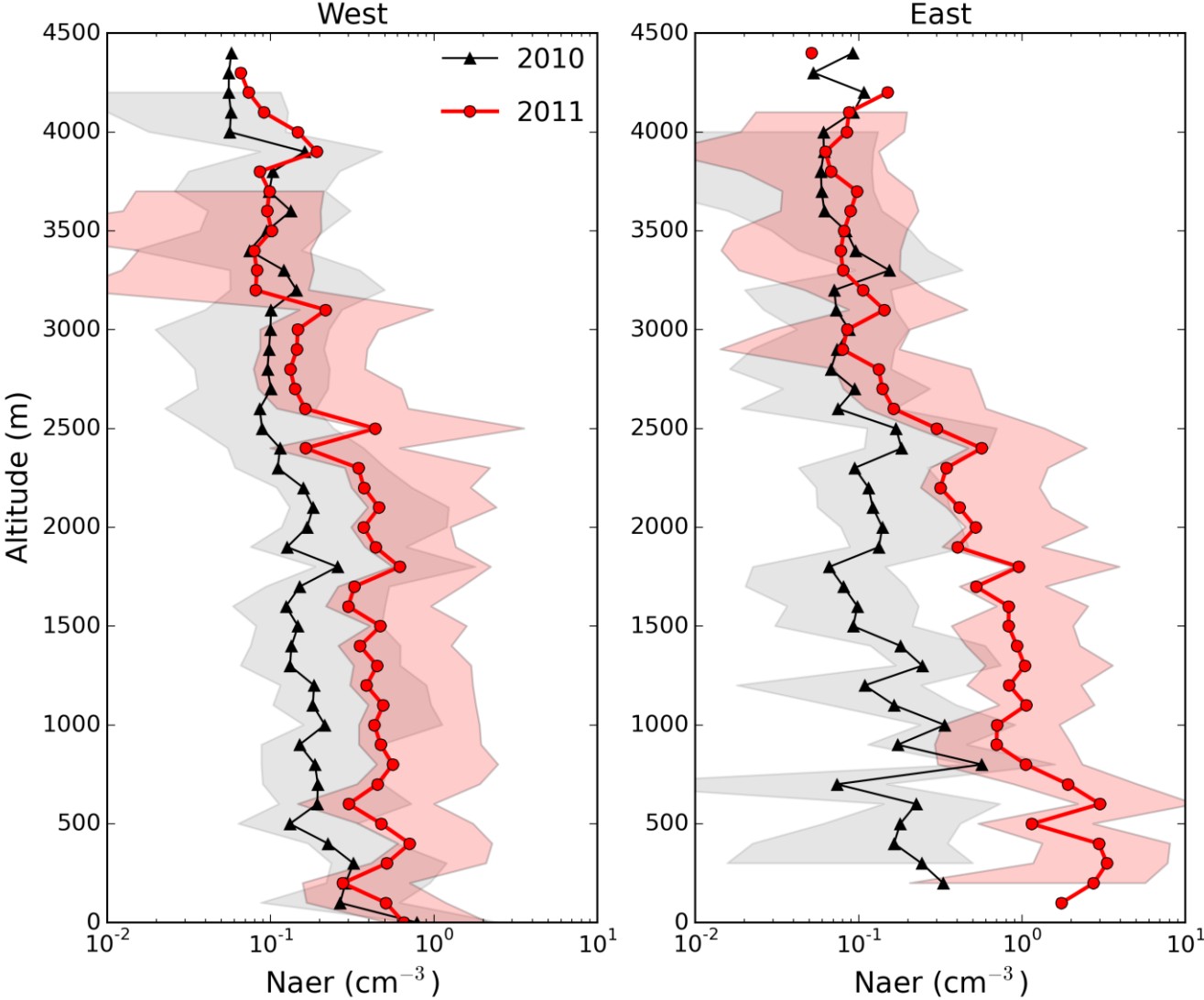

**Figure 13: Averaged vertical profiles of number concentration of aerosols (cm⁻³) on both sides of the Peninsula over all the flights (as for Figure 11). Shaded areas indicate the spread of the data between absolute minimum/maximum values at each level. West refers to 74-67°W and East refers to 65-60°W.**

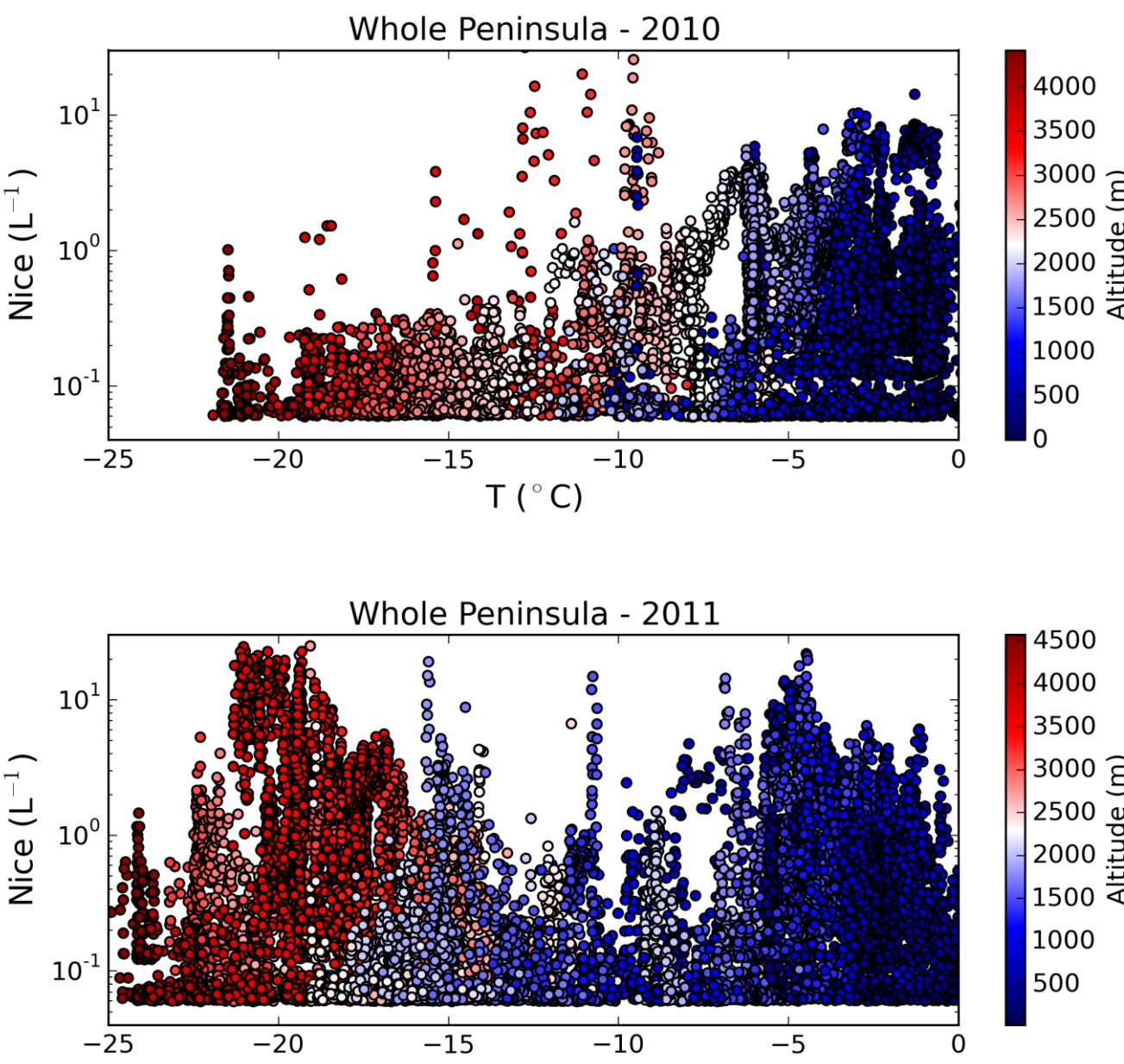

**Figure 14:** Distribution of all measurements of ice crystals (g$^{-1}$) as a function of atmospheric temperatures for 2010 (top) and 2011 (bottom) for the whole Peninsula. Colour indicates the altitude of the measurement with light to dark red referring to increasing altitudes above ≈2500 meters, and colours from light to dark blue to altitudes decreasing below ≈2250 meters.

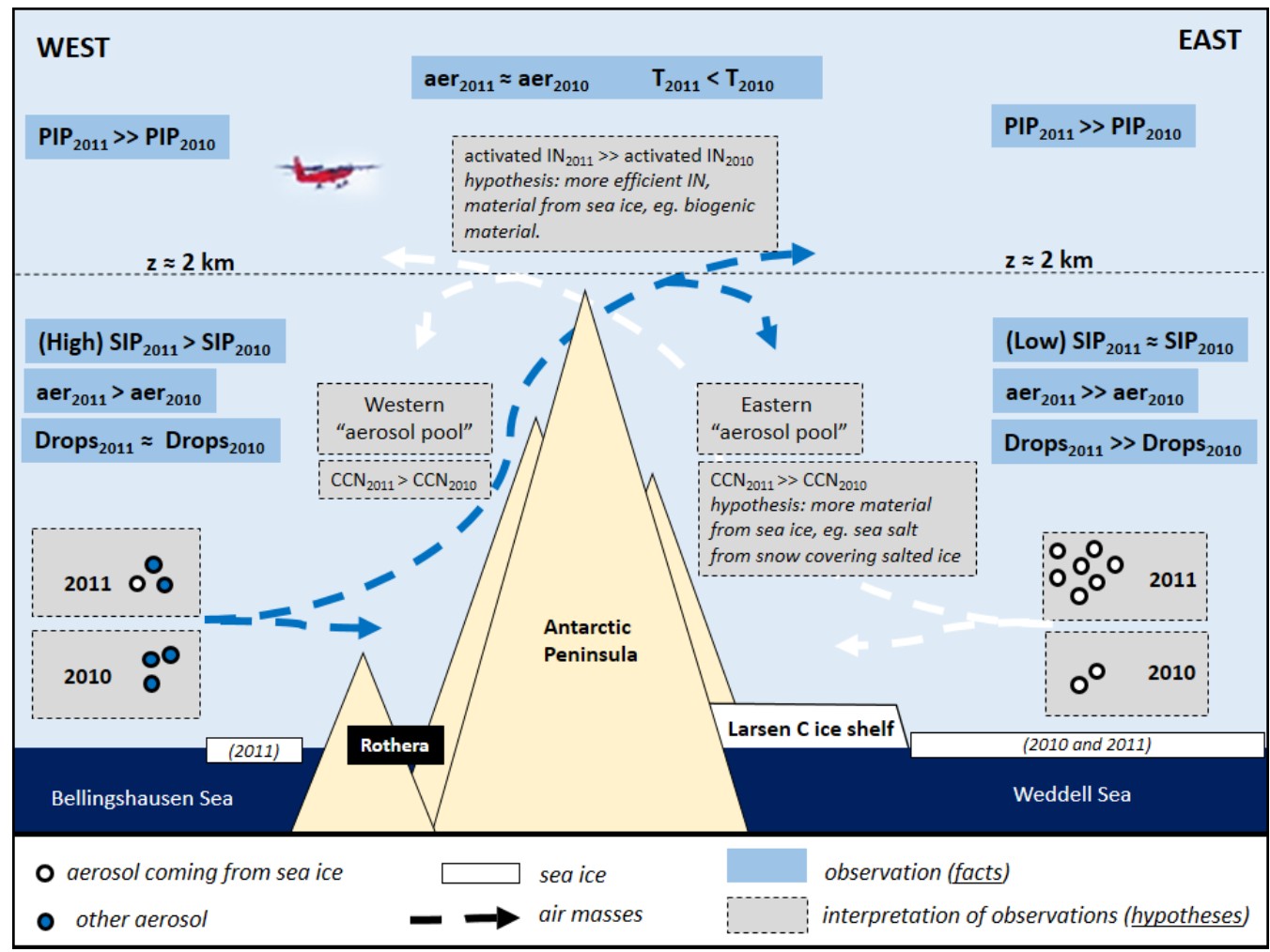

**Figure 15: Schematic summarising main observations from aircraft measurements from both periods of interests (February 2010 and January 2011) and on both sides of the Antarctic Peninsula (blue rectangles), as well as hypotheses (grey framed rectangles) proposed to explain observations, as presented in the discussions (section 4). PIP refers to Primary Ice Production while SIP refers to Secondary Ice Production, as presented in section 3.2.**