# Peer review of "The Microphysics of Clouds over the Antarctic Peninsula – Part 1: Observations"

_Atmospheric Chemistry and Physics, 2016_

## Short Comment (SC1) · 1 Jun 2016

The authors present an interesting and relevant analysis in their manuscript.

1) Reference to Grosvenor et al 2012 is incomplete.

2) This comment refers to the ice crystal numbers presented in Fig 7, and the authors' interpretation. They attribute number peaks at about -5°C to the Hallet-Mossop process (H-M) (after Hallet and Mossop, 1974).

My question is, how sure are the authors that the observations are due to the H-M process and not due to one of the other secondary ice formation processes, such as collision fragmentation (splinters produced by ice-ice collision, eg. Vardiman 1978, Takahashi 1995), droplet shattering (splinters produced during freezing of large droplets,

[Figure]

Leisner et al. 2014) or sublimation fragmentation (separation of ice particles from a parent ice particle when the connecting ice bridge sublimates, Bacon et al. 1998). Do any of their other measurements and observations (for example droplet diameter) support their assumption of the H-M process in preference to other secondary ice production processes?

Bacon, N. J., B. D. Swanson, M. B. Baker, and E. J. Davis, 1998: Breakup of levitated frost particles, J. Geophys. Res., 103(D12), 13763–13775, doi:10.1029/98JD01162. Leisner, T., T. Pander, P. Handmann, and A. Kiselev, 2014: Secondary ice processes upon heterogeneous freezing of cloud droplets. 14th Conf. on Cloud Physics and Atmospheric Radiation, Boston, MA, Amer. Meteor. Soc., 2.3 Mossop, S. C., 1985: Secondary ice particle production during rime growth: the effect of drop size distribution and rimer velocity.Q. J. R. Meteorol. Soc.,111, 1113-1124. Takahashi, T., Y. Nagao, and Y. Kushiyama, 1995: Possible High Ice Particle Production during Graupel–Graupel Collisions. Journal of the Atmospheric Sciences, 52,4523-4527. Vardiman, L., 1978: The generation of secondary ice particles in clouds by crystal-crystal collision. J. Atmos. Science, 35, 2168-2180.

---

## Referee Comment (RC1) · Anonymous Referee #1 · 25 Jun 2016

Review of "The microphysics of clouds over the Antarctic peninsula- Part I Observations" by Lachlan-Cope et al.

Recommendation: Requires major revision before acceptance for publication

This paper presents observations of clouds over the Antarctic peninsula for two seasons, and notes that there were substantial differences in the cloud properties between the two years because air masses in one of the years were more likely to have passed close to the surface over the sea ice in the Weddell Sea. Consequently, the authors claim that the sea ice covered Weddell Sea can act as a source of both cloud condensation nuclei and ice nuclei. I am definitely of the opinion that this paper should be published because there are very few observations in this region, and such observations are sorely needed in order to improve climate and numerical weather prediction

models in this area. However, I did not find that all of the analysis presented in the paper to be sufficiently thorough, and recommend some additional analysis and dependencies that should be examined before the paper should be accepted for final publication in ACP. My concerns are more thoroughly documented below.

Major Comments

1. There should be a better description of the limitations and uncertainties of the measurements. For example, is there any potential problem with shattering on the tube of the CAPS artificially amplifying concentrations of particles? Was any effort made to identify and remove shattered artifacts? These can be important even up to 500 microns. The authors state that they ignore particles smaller than 200 micrometer equivalent diameter because they make minimal contributions compared to the number measured by the CAS. This seems a bit counterintuitive because it suggests that there is a range between 50 and 200 microns where there were few particles. Because of the dependence of the CIP probe sample volume on diameter, especially for smaller particles, even if there are very few counts the concentrations of such particles can be large because of a small and poorly defined sample volume. Thus, I am not sure how robust the analysis of particles over this size range actually is. The authors stated that the phase identification scheme was tested by examining sorted images by eye. But, how well can you determine the phases of the smaller particles by eye, especially from the relatively coarse resolution of the photodiodes of the CIP? Information about the shape of the size distribution measured by the CAS (flatter indicative of ice, peaked indicative of supercooled liquid) could also have been used. Was there any Rosemount icing detector that could also have helped identify the phase.

2. It is mentioned that the CAS is used to estimate aerosol concentrations outside of clouds, and that all flights were used even when no clouds were present. But, it should be noted that if there is going to be any interpretation on how aerosols affect cloud properties, probably only the data from when clouds were detected should be used. The other flights might have very different meteorological conditions and hence

their analysis is not relevant. Secondly, has any efforts been made to determine if the humidity varied between flights? The humidity may have an impact on the aerosol concentration as the aerosols can expand as they are humidified. This may have an impact on the interpretation of the results. And, finally it is assumed that the measured aerosol concentration will "bear some relation to the number of CCN and IN available." However, there is no simple 1:1 relation because the CCN also depend on the super-saturation in cloud, which in turn will depend on the vertical velocity. Further, since only about 1 in 1 million aerosol particles is an IN, there is no guarantee there will be a strong relation between aerosol number and IN. Therefore, these assertions about aerosols should be better justified or the appropriate caveats should be added into the text.

3. Averaging cloud properties over one-degree longitude bands represents a very coarse average of quantities that can vary over much smaller scales. How much variability of the cloud properties were there within that one degree band? The authors state that "it would be expected that the variability observed within each individual cloud would be less than the variability between different clouds measured on different occasions." This might not necessarily be true. Can some analysis be performed or presented to show that this is actually the fact?

4. I think there needs to be a better description of the meteorological context of the observations. Given some of my comments above, I think there could be a strong dependence of both the cloud and aerosol properties on the wind direction and speed. How much variation in longitude bands exists between flights? It would be interesting to see if there was also dependence on these wind speeds, vertical velocities, lower atmospheric stability, etc. It may also be possible that the background concentrations of aerosols in the free troposphere could also be affecting the cloud properties. The dependence on meteorological conditions may be much greater than the differences between 2010 and 2011 that the authors are highlighting. Similarly, Table 2 that gives the significance of aerosol differences between years and regions does not imply causation. Could some of these differences be caused by varying meteorological conditions, varying relative humidity or cloud vertical velocities, varying horizontal velocities and wind directions, etc.? It may be possible to investigate some of these relationships better with the data available. It might be possible to get this information if case-by-case studies were performed in addition to such coarse averaging.

5. I think that the analysis on the cloud properties could be presented more thoroughly. Was there any dependence on how the cloud properties vary with height or temperature or with the distance from cloud top to where the observations were made. In addition to showing the cloud properties, is there any information available on the coverage of clouds or the frequency of occurrence of different phases? Is there any variation in how the cloud particle sizes vary in the different conditions?

6. There are many speculative comments in the manuscript that should be either better justified or removed. For example, on page 6 the authors state that a peak in ice number concentration around -5C "is probably related to a secondary ice production process....". While this statement may be true, it should be much better justified. For example, splinter production in the Hallett-Mossop process typically requires a 0.2 to 5 m/s impaction speed and the presence of droplets greater than 23 micrometer in diameter. Were such conditions present? Similarly, the discussion in the paragraph from lines 13 to 16 on page 6 is entirely speculative given that little information about the number of ice nuclei available at any temperature is available. In addition, for the analysis presented around line 30 on page 8, there needs to be presentation of other conditions (e.g., droplet sizes) to verify that the peak in ice crystal numbers is most likely due to the Hallett-Mossop process. I also feel that a lot of the discussion in the first full paragraph on page 9 is highly speculative and should be redrafted to clarify what is clearly known, and what are speculative comments regarding as to where primary and secondary production processes are occurring. Is it possible to present histograms or probability distribution functions of the different parameters to more clearly see the role of secondary ice crystal production processes? There are also many speculative

comments in the summary and conclusion that should be adjusted accordingly. There are too many words like "seems", "cloud", "might", etc.

7. Another place the speculative comments are present is from lines 9 to 13 on page 8. The authors state that it "is POSSIBLE that sea salt . . . could be lifted into the air as blowing snow. . ..and this has been SUGGESTED as an efficient mechanism for getting sea salt . . ..into the boundary layer. This SUGGESTS that the increase in droplet number concentrations . . . COULD results from the increase in aerosol numbers." Perhaps looking at the data on a case-by-case basis as well as looking at the averages would allow the authors to remove some of these highly speculative comments.

8. The analysis stratifies the data in a very averaged sense. In addition to this type of analysis, it would be very interesting to also stratify the observed cloud properties by cloud type since the sampling of types and altitudes may have varied on different flights and hence can affect some of the statistical analysis presented.

9. I recommend the use of ice nucleating particles (INPs) rather than ice nuclei (IN) to be consistent with currently excepted terminology.

Detailed Comments

1. Page 1, line 13: I would find the abstract more satisfying if significantly more was replaced with something more quantitative.

2. Page 1, line 25: I recommend labeling the locations of Palmer Station in Figure 1 as not all readers may be familiar with its location

3. Page 1, line 28: Recommend using the more standard terminology of ice nucleating particle (INP) rather than ice nuclei following the terminology adopted by Vali (2011).

4. Page 3, line 13: What does reasonably well mean quantitatively?

5. Page 3, line 22: Was any graupel present? It would seem that the circularity of graupel particles might approach those of liquid drops. In addition, how does the resolution

of the photodiodes affect the application of this scheme for smaller particles? Should the thresholds for circularity be dependent on the sizes of the particles?

6. Page 4, line 7: What is equivalent diameter? Is this an area equivalent diameter?

7. Page 4, line 12: What was the shape of the CAS size distributions? If it was peaked, this would offer more evidence that the particles were indeed supercooled water.

8. Page 5, line 22: suggest changing "are" to "were"

9. Page 5, line 23: suggest changing "are" to "were"

10. Page 5, line 26: Can slightly be quantified?

11. Page 8, line 19: Not enough evidence has been presented to definitively state that this is secondary ice production due to the Hallett-Mossop process.

12. Figures 3 and 4: The black shading is very faint and not be easy to see when the article is produced. Can you make the shading darker?

---

## Referee Comment (RC2) · Anonymous Referee #2 · 2 Jul 2016

This paper compares cloud properties measured East and West of the Antarctic Peninsula during 2010 and 2011. The authors find more water drops and ice crystals in 2011, particularly in the East. They suggest that this could be due to air masses passing over the sea ice in the Weddell Sea more frequently in 2011.

General Comments: This work represents a major contribution to our understanding of clouds near the Antarctic Peninsula – which is currently quite limited - and thus should be published. Reviewer 1 brings up a number of important points that would be very interesting to see explored, and which seem to be quite relevant to supporting the authors' hypothesis. However, the paper already has 13 figures and 3 tables. Given the title (". . . Part 1: Observations"), perhaps a reasonable option is to modify the manuscript to focus more on the observations themselves, adding more detail related to the cloud property measurements and the meteorological conditions dur-

ing the flights (as indicated by Reviewer 1), and relegate a detailed exploration of the hypothesis to Part II.

Major Comments:

1) Given the lack of knowledge of the properties of Antarctic clouds, this paper would benefit from an overall summary of the cloud properties that were measured, which would then be followed by the breakdown by year and location. A quantitative summary of the results (e.g. state the cloud properties measured) should also be included in the abstract.

2) Please clarify what data shows averages over flights and what shows averages of averages. For example, in Fig. 3, the points are averages over flights, correct? Why are there only 1 to 5 flights in each longitude bin / year when there were 12 flights?

3) The authors say on page 4, line 17: "the variability observed within each individual cloud would be much less than the variability between different clouds." Can they give a sense of what the variability within clouds and between clouds was? For example, it would be good to give the standard deviation for a few clouds, and for each flight within each longitudinal bin.

Minor comments:

1) Descriptions of the CIP and CAPS instrument are very brief, with references to other papers. More information should be given in the paper regarding these instruments and their accuracies.

2) Editing for grammar and clarity is needed throughout. Some examples: Page 2, line 15: "concludes on the possible implications" Page 2, line12: Before "In section 2" you should add an introductory sentence, e.g. "This paper is organized as follows." Page 6, line 21 "the CAPS probe will measure an aerosol . . ." is better stated "the CAPS probe measures aerosol . . ." Page 8, lines 2-6: Break up this sentence. Start a new sentence after "... in the east in 2011"

3) Tables 1 and 2: It would be helpful if the statistically significant numbers (or pairs of numbers in Table 1) were in bold font so they stand out.

---

## Author Comment (AC1) · 8 Sep 2016

1) We will correct this. 2) Of course we cannot be sure that the peak ice production around -5°C is due to the Hallet-Mossop process as we do not have the resolution (both temporal and spatial) to observe process in detail, we are only observing the result after the many small crystals formed in the H-M process have grown rapidly to a size that can be observed by the CIP. For this reason we said that the peak was probably due to H-M and did not definitely attribute the peak to any one process. The other mechanisms suggested in the comment all operate around -15°C (although the Bacon et al process could occur at warmer temperatures the process is a strong function of shape and is likely to be strongest with the dendrites that form at around -15°C) and so are unlikely to be responsible for the peak we see at -5°C. We do not know of any other mechanisms that have been reported to work at these warm temperatures. We

will add a sentence to the paper to make this clear.

---

## Author Response (AR1)

Reviewer 1 Major Comments

1)There should be a better description of the limitations and uncertainties of the measurements. For example, is there any potential problem with shattering on the tube of the CAPS artificially amplifying concentrations of particles? Was any effort made to identify and remove shattered artifacts? These can be important even up to
5  500 microns.

> Some extra text has been added (page 2 lines 14-16) on the effect of shattering in our CAPS probe. We have removed the shattered artifacts by observing the inter arrival time in the CIP. The physical modifications to the instrument to reduce shattering was the removal of the shroud on the CAS inlet

The authors state that they ignore particles smaller than 200 micrometer equivalent diameter because they
10  make minimal contributions compared to the number measured by the CAS.

>We ignore the particles in the CIP below 200 micrometers because we cannot tell whether they are ice or liquid drops for certain. We then go on to say that even if we count them all as drops they make a minimal contribution (2-3%) (added page 4 line 17)to those measured by the CAS so as far as the drops are concerned ignoring the small particles (in terms of numbers) is acceptable.

15  This seems a bit counterintuitive because it suggests that there is a range between 50 and 200 microns where there were few particles. Because of the dependence of the CIP probe sample volume on diameter, especially for smaller particles, even if there are very few counts the concentrations of such particles can be large because of a small and poorly defined sample volume.

> In a shadow probe like the CIP if we reject edge particles (ones that occult the ends of the CCD array) the
20  effective volume gets smaller as the particles get larger – we allow for this when calculating the particle concentrations. So it is for the larger particles that the sample volume becomes small and perhaps poorly defined. We have added some text to paper to make this clear (page 3 last sentence). As described previously we suggest that the droplet number concentration is dominated by particles smaller than 50um. As shown below, the CAS number concentration has a mode at 10 to 20 um and then decreases rapidly with increasing
25  size. The referee is correct that the CIP sample volume is also smaller for very small particles, meaning that it may overestimate the concentration of particles less than 200um due to miss sizing particles. However, this would mean that droplets in the 50 to 200 um size range would make a smaller contribution to the total concentration than the already minimal 2 to 3% we suggest as a maximum."

Thus, I am not sure how robust the analysis of particles over this size range actually is. The authors stated that
30  the phase identification scheme was tested by examining sorted images by eye. But, how well can you determine the phases of the smaller particles by eye, especially from the relatively coarse resolution of the photodiodes of the CIP? Information about the shape of the size distribution measured by the CAS (flatter indicative of ice, peaked indicative of supercooled liquid) could also have been used. Was there any Rosemount icing detector that could also have helped identify the phase?

35  >We have already stated that we cannot determine the phase of the smaller (we have used a size of 50 pixels as a cut off) particles in the CIP so we have not included them in the analysis. We know (see above)that this means

a relative small error in the number of drops. It is perhaps a larger error in the number of ice crystals – but we feel we cannot include these smaller CIP particles in with crystal count. We do see at all times peak in the supercooled water drops in the CAS – the averaged graph below (Size spectrum of CAS particles) gives some indication of this and we have included text (page 4 lines 21-22) indicating the presence of a peaked spectrum in the CAS in clouds.

[Figure]

2.It is mentioned that the CAS is used to estimate aerosol concentrations outside of clouds, and that all flights were used even when no clouds were present. But, it should be noted that if there is going to be any interpretation on how aerosols affect cloud properties, probably only the data from when clouds were detected should be used. The other flights might have very different meteorological conditions and hence their analysis is not relevant.

> We did not include flights where no clouds were present – in these locations it is very rare for there to be no clouds. The "non-cloud" flights were investigating either large scale flow over the peninsula or boundary layer fluxes over the sea ice. For the most part these flights tried to avoid clouds so that the nine hole turbulence probe did not get blocked with water. Although these flights may have occasionally entered clouds no attempt was made to make the cloud sampling representative of the cloud as a whole – for this reason these flights were excluded from the cloud analysis. There seemed to be no reason why they could not be included in an analysis of the aerosols. We have changed the text to make this clear (page 5 line 5). We have now added results from the 24 flights only to table 3 for comparison. We have also corrected some minor errors in the calculation of the values in table 3 – these corrections do not alter any of our conclusions.

Secondly, has any efforts been made to determine if the humidity varied between flights? The humidity may have an impact on the aerosol concentration as the aerosols can expand as they are humidified. This may have an impact on the interpretation of the results.

>We have now included a section (page 5 lines 13-21)on meteorological conditions during the campaign including the humidity. Of course the humidity outside clouds varies from flight to flight but there is no apparent correlation between the humidity and aerosol numbers.

And, finally it is assumed that the measured aerosol concentration will "bear some relation to the number of CCN and IN available." However, there is no simple 1:1 relation because the CCN also depend on the super-saturation in cloud, which in turn will depend on the vertical velocity.

>We understand that there is no simple 1:1 relationship between CCN and aerosol numbers and that is why we used the phrase "bear some relation to". As we do not fly a CCN counter this is the best we can do.

Further, since only about 1 in 1 million aerosol particles is an IN, there is no guarantee there will be a strong relation between aerosol number and IN. Therefore, these assertions about aerosols should be better justified or the appropriate caveats should be added into the text.

>The parameterisation of DeMott (2010) clearly show there is, at least in some data sets, a strong relation between aerosols greater than 0.5micron and INP. We have added the DeMott references to the paper to emphasize this point (page 5 line 10).

3. Averaging cloud properties over one-degree longitude bands represents a very coarse average of quantities that can vary over much smaller scales. How much variability of the cloud properties were there within that one degree band?

>We included the values for each individual flight on the longitude plots to show the variability between flights within a one degree band. On the whole  there are too few flights within each band to calculate standard deviations etc – which is why we have included the individual flight values and then calculate value for each side of the peninsula as a whole.

The authors state that "it would be expected that the variability observed within each individual cloud would be less than the variability between different clouds measured on different occasions." This might not necessarily be true. Can some analysis be performed or presented to show that this is actually the fact?

>In fact analysis shows that the variability is similar whether each individual data point is considered or if each individual flight is averaged – we have removed the sentence "it would be expected that the variability observed within each individual cloud would be less than the variability between different clouds measured on different occasions" as it isn't really relevant – the averaging of each flight was done to avoid long flights in a single bin dominating the overall average.

4. I think there needs to be a better description of the meteorological context of the observations.

>A section on the background meteorological conditions has been added to the paper (page 5 lines 12-21).

Given some of my comments above, I think there could be a strong dependence of both the cloud and aerosol properties on the wind direction and speed. How much variation in longitude bands exists between flights? It would be interesting to see if there was also dependence on these wind speeds, vertical velocities, lower atmospheric stability, etc. It may also be possible that the background concentrations of aerosols in the free troposphere could also be affecting the cloud properties. The dependence on meteorological conditions may be much greater than the differences between 2010 and 2011 that the authors are highlighting.

>Of course the aerosol concentration and properties will depend on wind direction. However, we believe just looking at the local wind direction close to the complex terrain of the Antarctic Peninsula will not tell us very much about the origin of the air mass the aircraft is flying through or the source (and hence properties) of the aerosol associated with the air mass. For this reason we have looked at back trajectories of air parcels from the aircraft position in section 4.1.

Similarly, Table 2 that gives the significance of aerosol differences between years and regions does not imply causation. Could some of these differences be caused by varying meteorological conditions, varying relative humidity or cloud vertical velocities, varying horizontal velocities and wind directions, etc.? It may be possible to investigate some of these relationships better with the data available. It might be possible to get this information if case-by-case studies were performed in addition to such coarse averaging.

> It would be interesting to look at particular cases and we plan to do this in future papers. The underlying purpose of this paper is to look – for the first time - at the average summer cloud properties and then use these average values to inform the model runs to be carried out in part 2 of this paper.

5. I think that the analysis on the cloud properties could be presented more thoroughly. Was there any dependence on how the cloud properties vary with height or temperature or with the distance from cloud top to where the observations were made?

>The clouds reported in this paper were almost all thin layer clouds and there was very little opportunity to take measurements at more than one level within the cloud the text has been slightly altered (page 3 line 2) to emphasize this. The variation of ice crystal number with temperature and height is already reported in figures 10 and 14.

In addition to showing the cloud properties, is there any information available on the coverage of clouds or the frequency of occurrence of different phases? Is there any variation in how the cloud particle sizes vary in the different conditions?

>We have not said anything about the coverage, frequency of clouds from the aircraft flights as of course these flights where made when clouds were present so any calculation of coverage would be biased. The cloud cover is the coastal areas of Antarctica are fairly extensive (see figure).

[Figure]

We have not considered the cloud particle sizes in any detail – this because most of the cloud we observed are rather thin layer clouds and we normally took measurements at one level. Of course we did make some vertical profiles and these showed the sort of variation expected (larger drops at the top of the cloud) however, this paper is trying to concentrate average spatial variability in the cloud across the Peninsula and we did not think we had enough observations in the vertical to say anything about this variation.

6. There are many speculative comments in the manuscript that should be either better justified or removed. For example, on page 6 the authors state that a peak in ice number concentration around -5C "is probably related to a secondary ice production process.". While this statement may be true, it should be much better justified. For example, splinter production in the Hallett-Mossop process typically requires a 0.2 to 5 m/s impaction speed and the presence of droplets greater than 23 micrometer in diameter. Were such conditions present?

>We have added an extra piece (page 6-7 lines 25 -13) to the paper describing why we think Hallet-Mossop process is the most likely for the secondary ice production. We, of course, have no information on the impact speed – this is something that could only be measured in the laboratory. However we have included the a discussion of the ratio of large drops to small drops that Mossop identified as a predictor of the Hallet-Mossop process.

 Similarly, the discussion in the paragraph from lines 13 to 16 on page 6 is entirely speculative given that little information about the number of ice nuclei available at any temperature is available.

>. The absence of a peak above -10degC on the east is clear in the figure. I assume the reviewer is referring to our assumption that the increase below -10degC is due to primary ice production. This seems to be an entirely reasonable assumption especially as it qualitatively agrees (within reason) with most commonly used

parameterisations with models that we show in figure 10. We have changed the text (page 7 lines 14-23) to emphasize this point.

In addition, for the analysis presented around line 30 on page 8, there needs to be presentation of other conditions (e.g., droplet sizes) to verify that the peak in ice crystal numbers is most likely due to the Hallett-Mossop process.

>We have included an explanation of why we believe the peak in ice production is due to the Hallett-Mossop process in section 3.2.

I also feel that a lot of the discussion in the first full paragraph on page 9 is highly speculative and should be redrafted to clarify what is clearly known, and what are speculative comments regarding as to where primary and secondary production processes are occurring.

>We have added text explaining why we believe primary and secondary ice production is occurring where we state in section 3 and we feel this makes it clear what we know and what are speculative comments.

Is it possible to present histograms or probability distribution functions of the different parameters to more clearly see the role of secondary ice crystal production processes?

>We already have included a large number of diagrams – including figures similar to histograms/probability distributions (eg figure 10 and 14)  - and we believe that further figures will not make role of secondary ice production clearer.

There are also many speculative comments in the summary and conclusion that should be adjusted accordingly. There are too many words like "seems", "cloud", "might", etc.

> See below dealing with similar issues.

7. Another place the speculative comments are present is from lines 9 to 13 on page 8. The authors state that it "is POSSIBLE that sea salt could be lifted into the air as blowing snow and this has been SUGGESTED as an efficient mechanism for getting sea salt into the boundary layer. This SUGGESTS that the increase in droplet number concentrations COULD results from the increase in aerosol numbers." Perhaps looking at the data on a case-by-case basis as well as looking at the averages would allow the authors to remove some of these highly speculative comments.

>Some of the comment are of course speculative  However, in absence of IN/CCN measurements, we believe that even case by case studies would remain speculative. However, the purpose of our discussion section is about discussing the possible explanations of statistically significant variations observed based on backtrajectories, and existing published literature. Indeed, We feel that these statements are justified as they are supported by the work of Yang et al (2008). This paper is specifically looking at the average values and there is would be no space for detailed case by case studies. Also the measurements made during this study did not include aircraft flight legs low over the sea ice (due to constraints in flight time from Rothera station) that would really be required to identify the exact source of the nuclei. We have taken recent measurements over the sea

ice in the Weddell sea and we hope these may shed some light on the source of nuclei – although this is not a simple matter.

8. The analysis stratifies the data in a very averaged sense. In addition to this type of analysis, it would be very interesting to also stratify the observed cloud properties by cloud type since the sampling of types and altitudes may have varied on different flights and hence can affect some of the statistical analysis presented.

>The clouds found at high southern latitudes are predominately thin layer clouds (often in multiple layers) and this was pointed out in section 2.1 – we have changed this very slightly (page 3 line 2) to make this clear. There would be no point in stratify the cloud properties by cloud type as there was on one type – stratus. We did look at the variation of properties with height and found that although properties did not change markedly with height the coverage of observations with height was too uneven to draw meaningful conclusions.

9. I recommend the use of ice nucleating particles (INPs) rather than ice nuclei (IN) to be consistent with currently excepted terminology.

>Changed

Detailed Comments

1. Page 1, line 13: I would find the abstract more satisfying if significantly more was replaced with something more quantitative.

>Something more quantitative added.

2. Page 1, line 25: I recommend labelling the locations of Palmer Station in Figure 1 as not all readers may be familiar with its location

> Location of Palmer Station added to text

3. Page 1, line 28: Recommend using the more standard terminology of ice nucleating particle (INP) rather than ice nuclei following the terminology adopted by Vali (2011).

>Changed

4. Page 3, line 13: What does reasonably well mean quantitatively?

>Value put for the agreement

5. Page 3, line 22: Was any graupel present? It would seem that the circularity of graupel particles might approach those of liquid drops. In addition, how does the resolution of the photodiodes affect the application of this scheme for smaller particles? Should the thresholds for circularity be dependent on the sizes of the particles?

>The scheme will have difficulty in identifying smaller particles and that is why we have used a 50 pixels as the lower cut off for particles identified in the CIP. This is equivalent to a circular drop of 200µm and this relatively

large size was chosen to make sure that most particles were correctly identified. No graupel was identified in any of the images looked at.

6. Page 4, line 7: What is equivalent diameter? Is this an area equivalent diameter?

>This has been made clear

7. Page 4, line 12: What was the shape of the CAS size distributions? If it was peaked, this would offer more evidence that the particles were indeed supercooled water.

>See the answer to major comment one above

8. Page 5, line 22: suggest changing "are" to "were"

>Changed

9. Page 5, line 23: suggest changing "are" to "were"

>Changed

10. Page 5, line 26: Can slightly be quantified?

>The numbers are shown in table 1

11. Page 8, line 19: Not enough evidence has been presented to definitively state that this is secondary ice production due to the Hallett-Mossop process.

>We feel we have now given enough evidence that the secondary ice production is due to the Hallett-Mossop process.

12. Figures 3 and 4: The black shading is very faint and not be easy to see when the article is produced. Can you make the shading darker?

>This has been changed

Anonymous Referee #2 comments

This paper compares cloud properties measured East and West of the Antarctic Peninsula during 2010 and 2011. The authors find more water drops and ice crystals in 2011, particularly in the East. They suggest that this could be due to air masses passing over the sea ice in the Weddell Sea more frequently in 2011.

5  General Comments:  This work represents a major contribution to our understanding of  clouds  near  the Antarctic  Peninsula  –  which  is  currently  quite  limited  -  and  thus should be published.  Reviewer 1 brings up a number of important points that would be very interesting to see explored, and which seem to be quite relevant to supporting the authors' hypothesis.  However, the paper already has 13 figures and 3 tables. Given the title ("Part 1:  Observations"),  perhaps a reasonable option is to modify the manuscript to focus more on

10  the observations themselves, adding more detail related to the cloud property measurements and the meteorological conditions during the flights (as indicated by Reviewer 1), and relegate a detailed exploration of the hypothesis to Part II.

>We have answered reviewers 1 points in some detail. We have added more information on the general climatic conditions during the flights. We feel that measurements here are limited – especially when compared

15  to measurements taken with larger better equipped aircraft – and we feel that we have extracted as much detail as we feel comfortable. Part 2 of this paper (at the moment in an advanced state of production) will concentrate on modelling the clouds during this period and trying to identify the best cloud scheme to use in forecasting and climate models.

Major Comments:

20  1) Given the lack of knowledge of the properties of Antarctic clouds, this paper would benefit from an overall summary of the cloud properties that were measured, which would then be followed by the breakdown by year and location. A quantitative summary of the results (e.g. state the cloud properties measured) should also be included in the abstract.

> Table 1 already gives a summary of the cloud properties measured broken down by year and location. We

25  have included a note of the measurements reported in the abstract and introduction.

2) Please clarify what data shows averages over flights and what shows averages of averages.  For example, in Fig.  3, the points are averages over flights, correct?

>Please note that the caption of figure 4 (note that the figure numbers have changed) makes it clear what is an average over a flight (the markers) and what is an average of the averages (the solid lines).

30  Why are there only 1 to 5 flights in each longitude bin / year when there were 12 flights?

> Each flight did not go to every longitude bin and even when it a flight visited a longitude bin it did not necessarily enter a cloud. For example for the bin nearest Rothera the temperature and humidity show a large number of points as most flights will have data in this bin while for the cloud parameter graphs there are less points as it was normal to avoid clouds while taking off and landing and flying close to the mountains.

3) The authors say on page 4, line 17: "the variability observed within each individual cloud would be much less than the variability between different clouds." Can they give a sense of what the variability within clouds and between clouds was? For example, it would be good to give the standard deviation for a few clouds, and for each flight within each longitudinal bin.

>We have removed this comment – the actual variability within clouds is similar to variability between clouds. Although this is interesting we did not want to add more data to a paper that already has a large numbers of figures and tables.

Minor comments:

1) Descriptions of the CIP and CAPS instrument are very brief, with references to other papers. More information should be given in the paper regarding these instruments and their accuracies.

>More detail of the CAPS instrument has been included (page page 3 lines 20-16)

2) Editing for grammar and clarity is needed throughout. Some examples: Page 2, line 15: "concludes on the possible implications" Page 2, line12: Before "In section 2" you should add an introductory sentence, e.g. "This paper is organized as follows." Page 6, line 21 "the CAPS probe will measure an aerosol" is better stated "the CAPS probe measures aerosol" Page 8, lines 2-6: Break up this sentence. Start a new sentence after "... in the east in 2011"

>These points (and others) have been corrected.

3) Tables 1 and 2: It would be helpful if the statistically significant numbers (or pairs of numbers in Table 1) were in bold font so they stand out.

>Figure in table 2 greater than 90% have been highlighted

[revised manuscript text omitted]

---

## Author Response (AR2)

2) Regarding Averaging. I think the source of confusion is Page 4, lines 24-27 through Page 5 lines 1-2. This paragraph is confusing and does not cover all kinds of data shown here. It seems to me there are 5 ways the data are shown (1) All data for all flights, as in Fig. 14. (2) Averages over each flight in longitudinal bins, shown as points in Fig. 4. (3) Averages of the flight averages in the bins, shown as lines in Fig. 4. (4) Averages of (1) over the entire East and the entire West for each year and (5) averages of vertical profiles for all flights for each year and side of the Peninsula, as in Fig. 13. I think they are each handled better as they come up, so I recommend omitting this entire paragraph except the sentence about in-cloud periods. The last two sentences could be moved into the first paragraph of Section 3.1.

> We have deleted the parts of this paragraph that are handled elsewhere – at the same making sure that the description of the averaging elsewhere is sufficient.

Also …

Page 5, lines 25 and 26, change "at each longitude" to "in each longitude bin."

>Corrected

Page 5, line 27-28, change "bins on each side" to "bins, one on each side" and change "one from one from 67 to 74˚W and the other from…" to just "from 67 to 74˚W and from …"

>Corrected

Regarding the authors' response: "Each flight did not go to every longitude bin and even when it a flight visited a longitude bin it did not necessarily enter a cloud. For example for the bin nearest Rothera the temperature and humidity show a large number of points as most flights will have data in this bin while for the cloud parameter graphs there are less points as it was normal to avoid clouds while taking off and landing and flying close to the mountains."

Why not include this information in the paper?

>This has been included in the paper (page 5 line 3)

Minor comments:

1) More detailed descriptions of the CIP and CAPs have been added, but the manuscript seems to go back and forth between them. I suggest reorganizing/rewriting the section on Page 3 Lines 11- 25 as follows. In addition, please rephrase, "although the hotwire sensor tends to under-read at high values of LWC" to indicate the implications. For example: "Part of this discrepancy is attributed to the hotwire sensor's tendency to under-read at high values of LWC."

The CAPS instrument contains three discrete instruments: The Cloud and Aerosol Spectrometer (CAS), the Cloud Imaging Probe (CIP), and the hotwire Liquid Water Content (LWC) sensor. Data from the hotwire LWC sensor was only used in this study to help validate the CAS data. The CAS and CIP are described in turn below.

5   The CAS measures the diameter of particles between 0.5 and 50 µm at a frequency of one Herz. While the CAS used in this campaign did not have a full anti-shatter inlet, modifications were made to reduce the effect of shattering on the inlet by removing the shroud that was originally fitted to the inlet. A previous study (Grosvenor et al., 2012), using a small subset of this data, reported errors with the data from the CAS instrument. In particular, it appeared to be over-counting when integrated water content from the CAS was
10  compared with measurements from the hotwire LWC sensor. After investigation, this was found to be due to air accelerating in the tube of the CAS instrument. Studies in the Cambridge University Markham wind tunnel using a fine pitot tube to measure the speed within the tube showed an increase corresponding to an increase in the count of 1.47; this has been accounted for in this latest study. When this correction is applied the LWC calculated by integrating the CAS data for most flights agrees to within 15% with the hotwire sensor – although
15  the hotwire sensor tends to under-read at high values of LWC.

The CIP images particles between a diameter of 25µm and 1.5mm, at a pixel resolution of 25 µm. While it had not at the time of this campaign been fitted with anti-shatter tips, a study of the particle inter-arrival times indicated very few shattered particles; these were removed by eliminating particles that arrived within 1 µs.
20
The CIP instrument produces shadow images …

>These suggestions have been included in the paper

25  2) There are still some small grammatical mistakes, but hopefully most can be fixed by the editors. A few suggestions for grammar/clarity are described here.

Page 2, line 29: Reads, "flights were made to study a variety of meteorological phenomena including boundary layer … and cloud studies." Change "cloud studies" to "cloud microphysical properties"

30  >We think cloud studies is more appropriate as although the flights mainly observed microphysical properties other parameters related to clouds (radiation etc.) were observed.

Page 3 line 6: add an apostrophe in "Survey" to make it "Survey's"

>Changed
35
Page 3 lines 9-10. Remove the sentence, "The CAPS probe … to 1.5 mm." since it is redundant with what follows.

>Changed

[revised manuscript text omitted]

10  CAS and the number of particles, seen by the CIP, less than 50 pixels (equivalent to a circular drop of 200μm diameter) are for all flights less than 2-3% of the number seen by the CAS and so we have ignored these particles whose phase we do not know. The version of the CAS probe used for this study was not able to measure polarization and so it was not possible to attempt to discriminate between solid and liquid with the CAS owever, it seems likely that the assumption that all CAS particles are liquid is  valid since as we find the liquid water calculated assuming the CAS particles are all

15  liquid agrees reasonably with that calculated from the hot wire probe on the CAPS (as stated above) Moreover we see a distinct peak (not shown) in the CAS size spectra, when in mixed phase clouds, indicative of drop formation.

In this study the average cloud properties over all the flights shown in Fig. 1 are considered. Each flight did not go to every longitude bin and even when it a flight visited a longitude bin it did not necessarily enter a cloud. For example for the bin nearest Rothera the temperature and humidity show a large number of points as most flights will have data in this bin. For the

20  cloud parameter graphs there are fewer points as it was normal to avoid clouds during take-off and landing and flying close to the mountains. ~~
[revised manuscript text omitted]